# PAC-Bayes Learning Bounds for
# Sample-Dependent Priors

**Pranjal Awasthi**
Google Research and
Rutgers University
pranjalawasthi@google.com

**Satyen Kale**
Google Research
satyenkale@google.com

**Stefani Karp**
Google Research and
Carnegie Mellon University
stefanik@google.com

**Mehryar Mohri**
Google Research and
Courant Institute of Mathematical Sciences
mohri@google.com

## Abstract

We present a series of new PAC-Bayes learning guarantees for randomized algorithms with sample-dependent priors. Our most general bounds make no assumption on the priors and are given in terms of certain covering numbers under the infinite-Rényi divergence and the $\ell_1$ distance. We show how to use these general bounds to derive learning bounds in the setting where the sample-dependent priors obey an infinite-Rényi divergence or $\ell_1$-distance sensitivity condition. We also provide a flexible framework for computing PAC-Bayes bounds, under certain stability assumptions on the sample-dependent priors, and show how to use this framework to give more refined bounds when the priors satisfy an infinite-Rényi divergence sensitivity condition.

## 1 Introduction

The PAC-Bayesian framework provides generalization bounds for the performance of randomized learning algorithms [McAllester, 1999b,a, Shawe-Taylor and Williamson, 1997]. Rather than outputting a single hypothesis, such algorithms output a probability distribution $Q$ (the posterior) over a hypothesis set $\mathcal{H}$. In the PAC-Bayes framework, the generalization guarantees associated with $Q$ are typically expressed in terms of the relative entropy, $\mathsf{D}(Q \parallel P)$, where $P$ is a fixed prior distribution over the hypothesis set. In the traditional framework, the prior $P$ must be selected before receiving a training sample [Langford and Caruana, 2002, Langford and Seeger, 2001, Seeger, 2002].

In recent years, there have been efforts to establish more refined PAC-Bayes bounds in which the prior can depend on the distribution generating the data or a separate sample drawn from the same distribution [Catoni, 2007, Ambroladze et al., 2007, Parrado-Hernández et al., 2012, Lever et al., 2013]. However, in practice, information about the underlying data distribution is available only via the training sample and discarding a fraction of that data to compute a generalization bound can be wasteful, motivating the study of PAC-Bayes bounds for *sample-dependent* priors.[1]

In the context of overparameterized deep neural networks, where deriving non-vacuous generalization bounds is notoriously hard, it has been argued that sample-dependent priors can lead to finer generalization bounds [Nagarajan and Kolter, 2019, Dziugaite and Roy, 2017, Neyshabur et al., 2018].

Sample-dependent priors can also lead to new learning methods. For instance, when training deep neural networks via stochastic gradient descent, a standard choice for the prior $P$ is a Gaussian centered around the parameters at random initialization. In this case, $D(Q \parallel P)$ is related to the distance from initialization of the final iterate's parameters. This, however, can be large in most realistic settings and it is therefore more appealing to choose as prior a Gaussian centered around parameters obtained by running some amount of pre-training using the training data, and subsequently use that prior as a guide for fine-tuning the parameters with additional training. This combination of pre-training followed by fine-tuning is common practice and clearly such a choice would be sample-dependent. Sample-dependent priors are also relevant in emerging scenarios such as adversarial training. A common practice here is to smooth a given classifier by injecting Gaussian noise into the inputs. This results in a classifier with a more favorable Lipschitz property, thereby improving robustness [Lecuyer et al., 2019, Cohen et al., 2019]. While the choice of the noise magnitude depends on the input, typically, these methods choose a priori a uniform noise magnitude across all inputs. It is much more appealing instead to choose a posterior over the noise magnitudes and inform this choice by carefully selecting a prior $P$ based on the sample, over the noise magnitudes, and using the prior $P$ as a regularizer to guide the search for the posterior.

From a theoretical perspective, there has been little work on generalization bounds for sample-dependent priors. The recent work of [Dziugaite and Roy, 2018a,b] took an important step in this direction by showing that for sample-dependent priors chosen via a differentially private mechanism PAC-Bayesian generalization bounds can be derived. They also showed that weaker conditions where the sample-dependent prior need only be "close" to a differentially private prior suffice for the bounds. We also recently became aware of [Rivasplata et al., 2020], which will appear at NeurIPS 2020 as well; this work also discusses general sample-dependent priors, although it is not yet apparent how the results compare. The following are our main contributions:

1. **General bounds via covering numbers**. We give general PAC-Bayes bounds, with no assumption on the sample-dependent priors, in terms of certain covering numbers of the priors. We provide two such bounds using covering numbers computed with the infinite-Rényi divergence and the $\ell_1$ distance.

2. **Bounds for stable priors**. We say that sample-dependent priors satisfy *prior stability*, if for any two samples $S$ and $S'$ that differ in exactly one input, the corresponding sample-dependent priors $P_S$ and $P_{S'}$ are close. *Closeness* here is measured either in terms of the infinite-Rényi divergence or the $\ell_1$ distance. For both cases, we show that our general covering-number-based bounds already give non-trivial generalization bounds.

3. **Framework for PAC-Bayes bounds under prior stability**. Building on the work of Foster et al. [2019] on *hypothesis set stability*, we provide a general method for deriving PAC-Bayes bounds assuming prior stability. We show how this method leads to refinements of the PAC-Bayes bound mentioned above for infinite-Rényi divergence prior stability.

**Related Work.** Our work builds on a strong line of work using algorithmic stability to derive generalization bounds, in particular [Bousquet and Elisseeff, 2002, Feldman and Vondrak, 2018, 2019, Bousquet et al., 2019]. Most significantly, our work builds on the recent notion of *hypothesis set stability* introduced by Foster et al. [2019].

We note that our work is not the first to combine PAC-Bayesian bounds and stability-like notions. Rivasplata et al. [2018] derive PAC-Bayesian bounds by randomizing the learned hypothesis output by a stable learning algorithm. However, their priors are only distribution-dependent (vs. sample-dependent), and they do not invoke any stability of the *priors*. London [2017] combines PAC-Bayes bounds and algorithmic stability, but remains in the setting of fixed, sample-independent priors.

The work of Dziugaite and Roy [2018a] is perhaps the most closely related to ours. Specifically, to our knowledge, Dziugaite and Roy [2018a] presents the first example of actually using the full $m$-item sample $S$ in order to generate a prior $P_S$. In particular, they assume that the priors are generated from samples via a *randomized* differentially-private mechanism. They then use results from the differential privacy literature (specifically [Dwork et al., 2015]) to show that, with high probability over both the choice of the sample *and* the sample-dependent prior, their PAC-Bayesian bounds hold with respect to this sample-dependent prior. In contrast, in this work we assume that priors are generated from samples in a *deterministic* manner, and furthermore, the mapping from samples to priors is *stable*, either in infinite-Rényi divergence or $\ell_1$ distance. In the case of infinite-Rényi divergence stable

priors, the priors themselves define a differentially-private mechanism for generating *hypotheses*. Thus, our setting is fundamentally different from that of Dziugaite and Roy [2018a].

## 2 Preliminaries

We use $\mathcal{X}$ and $\mathcal{Y}$ to denote the input and output spaces, respectively. For convenience, we define $\mathcal{Z} := \mathcal{X} \times \mathcal{Y}$, and denote by $\mathcal{D}$ a distribution over $\mathcal{Z}$ from which samples are drawn. We let $\mathcal{H}$ denote a hypothesis set of functions mapping from $\mathcal{X}$ to $\mathcal{Y}'$, and use $\Delta(\mathcal{H})$ throughout to denote the set of distributions on $\mathcal{H}$.

We consider a loss function $\ell \colon \mathcal{Y}' \times \mathcal{Y} \to [0, 1]$ and use $L(h, z)$ as shorthand to denote the composition $\ell(h(x), y)$. The expected loss of a randomized classifier parameterized by a distribution $Q \in \Delta(\mathcal{H})$ is the following expectation: $\mathbb{E}_{\substack{h \sim Q \\ z \sim \mathcal{D}}}[L(h, z)]$. For simplicity, we use $L_z$ to denote the vector $(L(h, z))_{h \in \mathcal{H}}$, allowing us to rewrite $\mathbb{E}_{\substack{h \sim Q \\ z \sim \mathcal{D}}}[L(h, z)]$ as $\mathbb{E}_{z \sim \mathcal{D}}\big[\langle Q, L_z \rangle\big]$.

Since the priors in this paper are sample-dependent, we denote by $P_S \in \Delta(\mathcal{H})$ a prior obtained after seeing the sample $S$. For two distributions $\mathcal{P}, \mathcal{Q}$ defined on the same discrete domain[2] $\Omega$, we use $\mathsf{D}(\mathcal{P} \parallel \mathcal{Q}) = \mathbb{E}_{\omega \sim \mathcal{P}}\left[\log\left(\frac{\mathcal{P}(\omega)}{\mathcal{Q}(\omega)}\right)\right]$ to denote the relative entropy (or KL divergence) of $\mathcal{P}$ from $\mathcal{Q}$, and we use $\mathsf{D}_\infty(\mathcal{P} \parallel \mathcal{Q})$ to denote the infinite-Rényi divergence (or max-divergence, as often seen in the differential privacy literature) of $\mathcal{P}$ from $\mathcal{Q}$, defined as follows: $\mathsf{D}_\infty(\mathcal{P} \parallel \mathcal{Q}) = \sup_{\omega \in \Omega} \log\left(\frac{\mathcal{P}(\omega)}{\mathcal{Q}(\omega)}\right)$. We will also need the notion of $\gamma$-approximate infinite-Rényi divergence, denoted $\mathsf{D}_\infty^\gamma(\mathcal{P} \parallel \mathcal{Q})$ for any two distributions $\mathcal{P}, \mathcal{Q}$: $\mathsf{D}_\infty^\gamma(\mathcal{P} \parallel \mathcal{Q}) := \sup_{A \subseteq \Omega \colon \mathcal{P}(A) \geq \gamma} \log\left(\frac{\mathcal{P}(A) - \gamma}{\mathcal{Q}(A)}\right)$. Finally, we use $\|\mathcal{P} - \mathcal{Q}\|_{\mathrm{TV}} = \frac{1}{2}\|\mathcal{P} - \mathcal{Q}'\|_1$ to denote the total variation distance between $\mathcal{P}$ and $\mathcal{Q}$.

Our bounds are stated in terms of Rademacher complexity, defined as follows. Let $S = (z_1, z_2, \ldots, z_m) \in \mathcal{Z}^m$ be a sample set sampled from $\mathcal{D}^m$, and let $\boldsymbol{\sigma} \in \{-1, 1\}^m$ be a vector of independent Rademacher variables. The notions of Rademacher complexity we need are:[3]

$$\widehat{\mathfrak{R}}_S(\mathcal{H}) = \frac{1}{m} \mathbb{E}_{\boldsymbol{\sigma}}\left[\sup_{h \in \mathcal{H}} \sum_{i=1}^m \sigma_i L(h, z_i)\right] \quad \text{and} \quad \mathfrak{R}_m(\mathcal{H}) = \mathbb{E}_S[\widehat{\mathfrak{R}}_S(\mathcal{H})].$$

## 3 General sample-dependent priors

In this section, we present general PAC-Bayes bounds for sample-dependent priors. Our bounds are in terms of certain covering numbers for sample-dependent priors, defined as follows.

**Definition 1.** *Let $\rho : \Delta(\mathcal{H}) \times \Delta(\mathcal{H}) \to \mathbb{R}$ be a divergence function taking values in non-negative reals. Let $m, n$ be positive integers. For a given sample $U$ of size $m+n$, and a scale $\alpha \geq 0$, $C \subseteq \Delta(\mathcal{H})$ is called a cover for $U$ at scale $\alpha$ under $\rho$ if for all subsamples $S \subseteq U$ with $|S| = m$, there exists a distribution $P \in C$ such that $\rho(P_S, P) \leq \alpha$. Define the covering number $\mathcal{N}(\alpha, m, U, \rho)$ to be the size of the smallest such cover. Define $\mathcal{N}(\alpha, m, n, \rho) = \max_{U \in \mathcal{Z}^{m+n}}[\mathcal{N}(\alpha, m, U, \rho)]$. When $m = n$, we use the notation $\mathcal{N}(\alpha, m, \rho)$ to mean $\mathcal{N}(\alpha, m, m, \rho)$.*

We now provide our general PAC-Bayes bounds with sample-dependent priors. These bounds are based on $\mathsf{D}_\infty$ and $\ell_1$ covering numbers, and the most general forms depend on two sample size parameters, $m$ and $n$. To keep the presentation clean, here we present the learning bounds using the $O(\cdot)$ notation for the special case $m = n$. The detailed bound without this assumption and proof can be found in Appendix A.1.

**Theorem 1.** *Let $P_S \in \Delta(\mathcal{H})$ be a prior over $\mathcal{H}$ determined by the choice of $S \in \mathcal{Z}^m$. Then, for any $\delta > 0$, with probability at least $1 - \delta$ over the draw of the sample $S \sim \mathcal{D}^m$, the following inequality holds for all $Q \in \Delta(\mathcal{H})$ and all $\alpha \geq 0$: if $D = \max\{\mathsf{D}(Q \parallel P_S), 2\}$,*

$$\mathbb{E}_{\substack{h \sim Q \\ z \sim \mathcal{D}}}[L(h, z)] \leq \mathbb{E}_{\substack{h \sim Q \\ z \sim S}}[L(h, z)] + O\left(\sqrt{\left(D + \alpha + \log \mathcal{N}(\alpha, m, \mathsf{D}_\infty) + \log(\tfrac{D}{\delta})\right)\left(\tfrac{1}{m}\right)}\right). \quad (1)$$

*Similarly, for any $\delta > 0$, with probability at least $1 - \delta$ over the draw of the sample $S \sim \mathcal{D}^m$, the following inequality holds for all $Q \in \Delta(\mathcal{H})$ and all $\alpha \geq 0$: if $D = \max\{\mathsf{D}(Q \parallel P_S), 2\}$,*

$$\mathop{\mathbb{E}}_{\substack{h \sim Q \\ z \sim \mathcal{D}}}[L(h,z)] \leq \mathop{\mathbb{E}}_{\substack{h \sim Q \\ z \sim S}}[L(h,z)] + O\left((\sqrt{D}+\alpha)\mathfrak{R}_{2m}(\mathcal{H}) + \sqrt{\left(\log\mathcal{N}(\alpha,m,\ell_1) + \log\left(\tfrac{D}{\delta}\right)\right)\left(\tfrac{1}{m}\right)}\right). \quad (2)$$

In order to establish the above theorem, we build upon the recently proposed framework of Foster et al. [2019] that provides generalization bounds for sample-dependent hypothesis sets. In particular, [Foster et al., 2019] consider a family of hypothesis sets $\mathcal{H} = (\mathcal{H}_S)_{S \in \mathcal{Z}^m}$ and show that generalization bounds for this family can be obtained via a notion of transductive Rademacher complexity. Formally, for a sample set $U$ of size $(m + n)$, define $\overline{\mathcal{H}}_{U,m} = \bigcup_{\substack{S \subset U, \\ |S|=m}} \mathcal{H}_S$. Then the transductive Rademacher complexity $\widehat{\mathfrak{R}}^{\diamond}_{U,m}(\mathcal{H})$ is defined for any $U = (z_1, \ldots, z_{m+n}) \in \mathcal{Z}^{m+n}$ as follows: if $\boldsymbol{\sigma}$ is a vector of $(m+n)$ independent random variables taking value $\frac{m+n}{n}$ with probability $\frac{n}{m+n}$ and value $-\frac{m+n}{m}$ with probability $\frac{m}{m+n}$, then

$$\widehat{\mathfrak{R}}^{\diamond}_{U,m}(\mathcal{H}) = \mathop{\mathbb{E}}_{\boldsymbol{\sigma}}\left[\sup_{h \in \overline{\mathcal{H}}_{U,m}} \frac{1}{m+n} \sum_{i=1}^{m+n} \sigma_i L(h, z_i)\right]. \quad (3)$$

Foster et al. [2019] gave the following generalization bound for $\mathcal{H}$ in terms of the maximum transductive Rademacher complexity, over all sample sets $U$ of size $m + n$.

**Theorem 2** (Theorem 1 in [Foster et al., 2019])**.** *Let $\mathcal{H} = (\mathcal{H}_S)_{S \in \mathcal{Z}^m}$ be a family of data-dependent hypothesis sets. Then, for any $\delta > 0$, with probability at least $1 - \delta$ over the choice of the draw of the sample $S \sim \mathcal{Z}^m$, the following inequality holds for all $h \in \mathcal{H}_S$:*

$$\mathop{\mathbb{E}}_{z \sim \mathcal{D}}[L(h,z)] \leq \mathop{\mathbb{E}}_{z \sim S}[L(h,z)] + 2 \max_{U \in \mathcal{Z}^{m+n}} \left[\widehat{\mathfrak{R}}^{\diamond}_{U,m}(\mathcal{H})\right] + 3\sqrt{\left(\tfrac{1}{m} + \tfrac{1}{n}\right)\log(\tfrac{2}{\delta})} + 2\sqrt{\left(\tfrac{1}{m} + \tfrac{1}{n}\right)^3 mn},$$

To apply the above result in our setting, recall that we interpret any distribution $Q \in \Delta(\mathcal{H})$ as a randomized hypothesis whose loss on any given point $z \in \mathcal{Z}$ is $\langle Q, L_z \rangle$. For a given $\mu > 0$, we then apply Theorem 2 to the following family of sample-dependent (randomized) hypothesis sets $\mathcal{Q}_{m,\mu} = (\mathcal{Q}_{S,\mu})_{S \in \mathcal{Z}^m}$ as

$$\mathcal{Q}_{S,\mu} = \left\{Q \in \Delta(\mathcal{H}) : \mathsf{D}(Q \parallel P_S) \leq \mu\right\}. \quad (4)$$

To apply Theorem 2, we need to bound the transductive Rademacher complexity of this family, $\widehat{\mathfrak{R}}^{\diamond}_{U,m}(\mathcal{Q}_{m,\mu}) = \mathbb{E}_{\boldsymbol{\sigma}}\left[\sup_{Q \in \overline{\mathcal{Q}}_{U,m,\mu}} \frac{1}{m+n} \sum_{i=1}^{m+n} \sigma_i \langle Q, L_{z_i} \rangle\right]$, where $\boldsymbol{\sigma}$ is a vector of random variables as defined just before (3). We establish such upper bounds (Lemmas 1 and 2 below) via the covering numbers from Definition 1, which leads to a bound similar to that of Theorem 1 in terms of $\mu$. The following lemma, proved in Appendix A.2, bounds the transductive Rademacher complexity using $\mathsf{D}_\infty$-covering numbers:

**Lemma 1.** *For any $\alpha \geq 0$, we have*

$$\widehat{\mathfrak{R}}^{\diamond}_{U,m}(\mathcal{Q}_{m,\mu}) \leq \sqrt{\left(\frac{\mu + \alpha + \log\mathcal{N}(\alpha, m, U, \mathsf{D}_\infty)}{2}\right)\left(\frac{1}{m} + \frac{1}{n}\right)^3 mn}.$$

We now give a bound (proved in Appendix A.3) in terms of $\ell_1$-covering numbers using a bit of notation. Let $m, n$ be two positive integers, and let $U = (z_1, z_2, \ldots, z_{m+n}) \in \mathcal{Z}^{m+n}$ be a sample set. Then we define a notion of Rademacher complexity $\tilde{\mathfrak{R}}_{U,m}(\mathcal{H})$ as follows: if $\boldsymbol{\sigma}$ is a vector of $(m+n)$ independent random variables taking value $\frac{m+n}{n}$ with probability $\frac{n}{m+n}$ and value $-\frac{m+n}{m}$ with probability $\frac{m}{m+n}$, then

$$\tilde{\mathfrak{R}}_{U,m}(\mathcal{H}) := \frac{1}{m+n} \mathop{\mathbb{E}}_{\boldsymbol{\sigma}}\left[\sup_{h \in \mathcal{H}} \left|\sum_{i=1}^{m+n} \sigma_i L(h, z_i)\right|\right]. \quad (5)$$

**Lemma 2.** *For any $\alpha \geq 0$, we have*

$$\widehat{\mathfrak{R}}^{\diamond}_{U,m}(\mathcal{Q}_{m,\mu}) \leq (\sqrt{2\mu} + \alpha)\tilde{\mathfrak{R}}_{U,m}(\mathcal{H}) + \sqrt{\frac{\log\mathcal{N}(\alpha, m, U, \ell_1)}{2}\left(\frac{1}{m} + \frac{1}{n}\right)^3 mn}.$$

We obtain a uniform bound (proved in Appendix A.4) over all values of $\mu$ by using a standard doubling argument:

**Lemma 3.** *Suppose the following bound holds with probability at least $1 - \delta$ over the choice of $S$: for all $Q \in \mathcal{Q}_{S,\mu}$,*

$$\mathbb{E}_{\substack{h \sim Q \\ z \sim \mathcal{D}}}[L(h,z)] \leq \mathbb{E}_{\substack{h \sim Q \\ z \sim S}}[L(h,z)] + f(\mu) + g(\delta),$$

*where $f$ is an increasing function of $\mu$ and $g$ is a decreasing function of $\delta$. Then, the following holds with probability at least $1 - \delta$ for all $Q \in \Delta(\mathcal{H})$:*

$$\mathbb{E}_{\substack{h \sim Q \\ z \sim \mathcal{D}}}[L(h,z)] \leq \mathbb{E}_{\substack{h \sim Q \\ z \sim S}}[L(h,z)] + f\big(2\max\{D(Q \parallel P_S),2\}\big) + g\left(\tfrac{\delta}{\max\{D(Q\|P_S),2\}}\right).$$

## 4   Stable sample-dependent priors

We now provide PAC-Bayes bounds for the setting where the sample-dependent prior $P_S$ satisfies a sensitivity assumption; i.e., for two samples $S$ and $S'$ of size $m$ that differ in only a single data point, the priors $P_S$ and $P_{S'}$ are close in some divergence defined on the pair of distributions over hypotheses. The precise definition of sensitivity follows.

**Definition 2.** *Let $\rho : \Delta(\mathcal{H}) \times \Delta(\mathcal{H}) \to \mathbb{R}$ be a divergence function taking values in non-negative reals. The family of sample-dependent priors $(P_S)_{S \in \mathcal{Z}^m}$ is said to have sensitivity $\epsilon$ w.r.t. $\rho$ if for all samples $S, S' \in \mathcal{Z}^m$ differing in a single data point, $\rho(P_S, P_{S'}) \leq \epsilon$.*

The specific divergences we will consider are the infinite Rényi divergence $\mathsf{D}_\infty$ and the $\ell_1$ distance. The bounds of Theorem 1 imply the following learning bounds under assumptions of $\mathsf{D}_\infty$ and $\ell_1$ sensitivity:

**Corollary 1.** *Suppose the family of sample-dependent priors $(P_S)_{S \in \mathcal{Z}^m}$ has $\mathsf{D}_\infty$ sensitivity $\epsilon$. Then, for any $\delta > 0$, with probability at least $1 - \delta$ over the draw of the sample $S \sim \mathcal{D}^m$, the following inequality holds for all $Q \in \Delta(\mathcal{H})$: if $D = \max\{\mathsf{D}(Q \parallel P_S), 2\}$,*

$$\mathbb{E}_{\substack{h \sim Q \\ z \sim \mathcal{D}}}[L(h,z)] \leq \mathbb{E}_{\substack{h \sim Q \\ z \sim S}}[L(h,z)] + O\left(\sqrt{\tfrac{D}{m}} + \epsilon + \log\left(\tfrac{D}{\delta}\right)\tfrac{1}{m}\right). \tag{6}$$

*Suppose instead that the family of sample-dependent priors $(P_S)_{S \in \mathcal{Z}^m}$ has $\ell_1$ sensitivity $\epsilon$. Then, for any $\delta > 0$, with probability at least $1 - \delta$ over the draw of the sample $S \sim \mathcal{D}^m$, the following inequality holds for all $Q \in \Delta(\mathcal{H})$: if $D = \max\{\mathsf{D}(Q \parallel P_S), 2\}$,*

$$\mathbb{E}_{\substack{h \sim Q \\ z \sim \mathcal{D}}}[L(h,z)] \leq \mathbb{E}_{\substack{h \sim Q \\ z \sim S}}[L(h,z)] + O\left((\sqrt{D} + \epsilon m)\mathfrak{R}_{2m}(\mathcal{H}) + \sqrt{\log\left(\tfrac{D}{\delta}\right)\tfrac{1}{m}}\right). \tag{7}$$

*Proof.* Suppose the family of sample-dependent priors $(P_S)_{S \in \mathcal{Z}^m}$ has $\mathsf{D}_\infty$ sensitivity $\epsilon$. Let $U \in \mathcal{Z}^{2m}$, and let $S$ be an arbitrary subset of $U$ of size $m$. It is then easy to see that $\{P_S\}$ is a cover for $U$ at scale $\epsilon m$ under $\mathsf{D}_\infty$, and (6) follows by immediately by applying (1) from Theorem 1. The bound for the $\ell_1$ case is exactly analogous. $\qquad\square$

We can obtain more nuanced bounds than the ones in Corollary 1 by exploiting the sensitivity of the priors via the concept of *hypothesis set stability* from [Foster et al., 2019]. In order to obtain PAC-Bayesian learning bounds using this framework, we first define several quantities in terms of a *family* of sample-dependent sets of distributions $\mathcal{Q}_m = (\mathcal{Q}_S)_{S \in \mathcal{Z}^m}, \mathcal{Q}_S \subseteq \Delta(\mathcal{H})$. This construction is analogous to (4); the only difference is that we have temporarily dropped $\mu$ for now, to emphasize that the following definitions are applicable to a general sample-dependent family. Specifically, we will assume that the family $\mathcal{Q}_m$ satisfies a certain *stability* property defined below:

**Definition 3.** *We say that $\mathcal{Q}_m = (\mathcal{Q}_S)_{S \in \mathcal{Z}^m}$ is $\beta$-uniformly stable for some $\beta \geq 0$ if $\forall S, S' \in \mathcal{Z}^m$ differing by exactly one point, for every $Q \in \mathcal{Q}_S$, there exists a $Q' \in \mathcal{Q}_{S'}$ such that $\|Q - Q'\|_{\mathrm{TV}} \leq \beta$.*

To describe our learning bounds, we need a bit of notation from [Foster et al., 2019]. We denote by $\boldsymbol{\sigma} \in \{-1, 1\}^m$ a vector of independent Rademacher variables. For two samples $S, T \in \mathcal{Z}^m$, we denote by $S_T^{\boldsymbol{\sigma}}$ the sample obtained from $S$ by replacing the $i$-th element of $S$ by the corresponding element

of $T$ for all $i$ such that $\boldsymbol{\sigma}_i = -1$. Finally, we define the following notion of Rademacher complexity for a family of sample-dependent sets of distributions $\mathcal{Q}_m = (\mathcal{Q}_S)_{S\in\mathcal{Z}^m}$:

$$\mathfrak{R}_m^\diamond(\mathcal{Q}_m) = \frac{1}{m} \mathop{\mathbb{E}}_{S,T,\boldsymbol{\sigma}} \left[ \sup_{Q\in\mathcal{Q}_{S_T^{\boldsymbol{\sigma}}}} \sum_{i=1}^m \sigma_i \langle Q, L_{z_i}\rangle \right], \tag{8}$$

where $z_i$ is element $i$ of sample $T$. With these definitions, we have the following learning bound. This is analogous to a bound from [Foster et al., 2019] and proven using similar techniques, but it is tighter because of the Rademacher complexity term multiplying the stability term in the bound. The proof appears in Appendix B.1.

**Theorem 3.** *Suppose $\mathcal{Q}_m = (\mathcal{Q}_S)_{S\in\mathcal{Z}^m}$ is $\beta$-uniformly stable. Then, for any $\delta > 0$, with probability at least $1 - \delta$ over the draw of the sample $S \sim \mathcal{D}^m$, the following holds for all $Q \in \mathcal{Q}_S$:*

$$\mathop{\mathbb{E}}_{\substack{h\sim Q\\z\sim\mathcal{D}}}[L(h,z)] \leq \mathop{\mathbb{E}}_{h\sim Q}\left[\frac{1}{m}\sum_{i=1}^m L(h,z_i)\right] + O\left(\mathfrak{R}_m^\diamond(\mathcal{Q}_m) + (1+\beta\mathfrak{R}_m(\mathcal{H})m)\sqrt{\tfrac{1}{m}\log(\tfrac{1}{\delta})} + \beta\log(\tfrac{m}{\delta})\right).$$

The proof of the theorem above is along the lines of the proof of Theorem 2 in [Foster et al., 2019]. Specifically, for two samples $S, S' \in \mathcal{Z}^m$, define the function $\Psi(S, S')$ as follows:

$$\Psi(S,S') = \sup_{Q\in\mathcal{Q}_S} \langle Q, \ell\rangle - \langle Q, \hat{\ell}_{S'}\rangle,$$

where $\ell, \hat{\ell}_{S'} \in \mathbb{R}^{\mathcal{H}}$ defined as $\ell(h) = \mathbb{E}_{z\sim\mathcal{D}}[L(h,z)]$ and $\hat{\ell}_{S'}(h) = \mathbb{E}_{z\sim S'}[L(h,z)]$, where $z \sim S'$ indicates uniform sampling from $S'$. The proof of the bound consists of applying McDiarmid's inequality to $\Psi(S,S)$. To do this, we need to analyze the sensitivity of this function, i.e., compute a bound on $|\Psi(S,S) - \Psi(S',S')|$ where $S'$ is a sample differing from $S$ in exactly one point. We provide a refined bound on the sensitivity using the fact that the map $Q \mapsto \langle Q, \ell\rangle$ is linear.

This bound leads to refined PAC-Bayesian bounds via the following template. (1) We define an appropriate sample-dependent family of distributions $\mathcal{Q}_m = (\mathcal{Q}_S)_{S\in\mathcal{Z}^m}$. Typically $\mathcal{Q}_S$ will be set to $\mathcal{Q}_{S,\mu} := \{Q \in \Delta(\mathcal{H}): \mathsf{D}(Q \parallel P_S) \leq \mu\}$ for some parameter $\mu$, as in (4). (2) Then, assuming that the priors $P_S$ are chosen to have $\epsilon$ sensitivity, we show that $\mathcal{Q}_m$ is $\beta$-stable for some small $\beta$ depending on $\epsilon$. (3) We also derive bounds on the Rademacher complexity $\mathfrak{R}_m^\diamond(\mathcal{Q}_m)$ in terms of $\mu$ and $\epsilon$. Using these bounds in Theorem 3 gives us a learning bound that depends on $\mu$. (4) We obtain a uniform bound over all possible values of $\mu$ via a standard union bound argument (Lemma 3).

We now proceed to instantiate this template for $\mathsf{D}_\infty$-sensitive priors. This leads to a better bound than the one in Corollary 1, with an extra assumption. The following theorem (precise bound spelled out in Appendix B.2) shows how to apply Theorem 3 to obtain refined PAC-Bayes bounds.

**Theorem 4.** *Suppose the family of sample-dependent priors $(P_S)_{S\in\mathcal{Z}^m}$ has $\mathsf{D}_\infty$ sensitivity $\epsilon$. Also assume that for some $\eta > 0$, we have $P_S(h) \geq \eta$ for all $h \in \mathcal{H}$, and all $S \in \mathcal{Z}^m$. Then, for any $\delta > 0$, with probability at least $1 - \delta$ over the draw of the sample $S \sim \mathcal{D}^m$, the following inequality holds for all $Q \in \Delta(\mathcal{H})$: if $D = \max\{\mathsf{D}(Q \parallel P_S), 2\}$,*

$$\mathop{\mathbb{E}}_{\substack{h\sim Q\\z\sim\mathcal{D}}}[L(h,z)] \leq \mathop{\mathbb{E}}_{h\sim Q}\left[\frac{1}{m}\sum_{i=1}^m L(h,z_i)\right] + O\Bigg(\sqrt{\frac{D}{m}} + \epsilon^2 + \epsilon\sqrt{\frac{\log(m/\eta)}{m}}$$

$$+ (1+\epsilon\mathfrak{R}_m(\mathcal{H})m)\sqrt{\tfrac{1}{m}\log(\tfrac{D}{\delta})} + \epsilon\log(\tfrac{mD}{\delta})\Bigg).$$

*Proof.* Define a sample-dependent family of distributions $\mathcal{Q}_m = (\mathcal{Q}_S)_{S\in\mathcal{Z}^m}$ where $\mathcal{Q}_S = \{Q: \mathsf{D}_\infty(Q \parallel P_S) \leq \mu\}$ for some parameter $\mu$. We now apply the bound in Theorem 3, using the bound on the Rademacher complexity from Lemma 4, and the bound $\beta \leq 2\epsilon$ from Lemma 6. Finally, a uniform bound over all values of $\mu$ follows by an application of Lemma 3. $\qquad\square$

At first glance, Theorem 4 may look qualitatively similar to Theorem 4.2 of [Dziugaite and Roy, 2018a]. Indeed, both bounds make use of the same tools developed in the differential privacy literature (specifically, [Dwork et al., 2015]), but the two analyses are completely different, since the settings are fundamentally different, as stated in the Introduction. This also makes the results incomparable.

The following is the key technical lemma needed in the proof of Theorem 4 to bound the Rademacher complexity term from Theorem 3 which employs the tools from [Dwork et al., 2015].

**Lemma 4.** *If* $\mathsf{D}_\infty(P_S \parallel P_{S'}) \le \epsilon$ *for all* $S, S' \in \mathcal{Z}^m$ *differing by exactly one point, and for some* $\eta > 0$, *we have* $P_S(h) \ge \eta$ *for all* $h \in \mathcal{H}$, *and all* $S \in \mathcal{Z}^m$. *Then*

$$\mathfrak{R}_m^\diamond(\mathcal{Q}_{m,\mu}) \le \sqrt{\frac{2\mu}{m} + 2\epsilon^2 + 2\epsilon\sqrt{\frac{\log(2m^2/\eta)}{m}}} + \sqrt{\frac{2}{m}} + \frac{1}{m}.$$

*Proof.* Fix the value of $\boldsymbol{\sigma}$. Consider the distribution $\mathcal{P}_{\boldsymbol{\sigma}}$ on $(S, T, h)$ induced by sampling $S, T \sim \mathcal{D}^m$, and then conditioned on the values of $S$ and $T$, sampling $h \sim P_{S_T^\sigma}$. Consider the marginal distribution of $h$ induced by $\mathcal{P}_{\boldsymbol{\sigma}}$. The probability assigned to $h$ in this marginal distribution is

$$\mathop{\mathbb{E}}_{S,T \sim \mathcal{D}^m}[P_{S_T^\sigma}(h)] = \mathop{\mathbb{E}}_{S \sim \mathcal{D}^m}[P_S(h)],$$

by symmetry, so this marginal distribution is independent of $\boldsymbol{\sigma}$. Call the marginal distribution $\mathcal{P}$.

Since sampling $h \sim P_{S_T^\sigma}$ is $\epsilon$-differentially private, by Theorem 20 in [Dwork et al., 2015], for any $\gamma > 0$, we have[4] $\mathsf{D}_\infty^\gamma(\mathcal{P}_{\boldsymbol{\sigma}} \parallel \mathcal{D}^{2m} \otimes \mathcal{P}) \le \kappa := \epsilon^2 m + \epsilon\sqrt{m \log(2/\gamma)}$.

Thus, by Lemma 9 in Appendix B.3 (which follows Lemma 3.17, part 1, in [Dwork and Roth, 2014]), there exists a distribution $\mathcal{P}'_{\boldsymbol{\sigma}}$ on $(S, T, h)$ s.t. (1) $\|\mathcal{P}'_{\boldsymbol{\sigma}} - \mathcal{P}_{\boldsymbol{\sigma}}\|_{\mathrm{TV}} \le \gamma$, (2) $\mathsf{D}_\infty(\mathcal{P}'_{\boldsymbol{\sigma}} \parallel \mathcal{D}^{2m} \otimes \mathcal{P}) \le \kappa$, and (3) the marginal distribution of $\mathcal{P}'_{\boldsymbol{\sigma}}$ on $(S, T)$ is $\mathcal{D}^{2m}$. Let $\mathcal{P}'_{\boldsymbol{\sigma}|S,T}$ denote the distribution of $h$ conditioned on $S$ and $T$, yielding $\mathbb{E}_{(S,T)\sim\mathcal{D}^{2m}}\left[\|\mathcal{P}'_{\boldsymbol{\sigma}|S,T} - \mathcal{P}_{S_T^\sigma}\|_{\mathrm{TV}}\right] = \|\mathcal{P}'_{\boldsymbol{\sigma}} - \mathcal{P}_{\boldsymbol{\sigma}}\|_{\mathrm{TV}} \le \gamma$, by (1) and (3). Then by Markov's inequality, with probability at least $\left(1 - \frac{1}{m}\right)$ over $(S, T) \sim \mathcal{D}^{2m}$, we have

$$\|\mathcal{P}'_{\boldsymbol{\sigma}|S,T} - \mathcal{P}_{S_T^\sigma}\|_{\mathrm{TV}} \le m\gamma. \tag{9}$$

Set $\gamma = \frac{\eta}{m^2}$ and assume $m \ge 2$ (the lemma statement holds trivially if $m = 1$). We use the shorthand $u_{\boldsymbol{\sigma}}(h) = \sum_{i=1}^m \sigma_i L(h, z_i)$, where $z_i$ is element $i$ of sample $T$, so that $\sum_{i=1}^m \sigma_i \langle Q, L_{z_i} \rangle = \langle Q, u_{\boldsymbol{\sigma}} \rangle$.

For any pair $(S, T)$ satisfying (9), we have the lower bound: $\mathcal{P}'_{\boldsymbol{\sigma}|S,T}(h) \ge \eta - m\gamma$ for all $h \in \mathcal{H}$. When (9) holds, we can therefore apply Lemma 5 with $\epsilon = 2m\gamma$ and $d_\infty = \eta - m\gamma$ and conclude: $\forall Q$ s.t. $\mathsf{D}(Q \parallel P_{S_T^\sigma}) \le \mu$, $\exists Q'$ s.t. $\mathsf{D}(Q' \parallel \mathcal{P}'_{\boldsymbol{\sigma}|S,T}) \le \mu$ and $\|Q - Q'\|_{\mathrm{TV}} \le \sqrt{\frac{\epsilon d_\infty}{2}} = \sqrt{\frac{m\gamma}{\eta - m\gamma}}$. Thus,

$$\sup_{\mathsf{D}(Q\|P_{S_T^\sigma})\le\mu} \langle Q, u_{\boldsymbol{\sigma}} \rangle - \sup_{\mathsf{D}(Q'\|\mathcal{P}'_{\boldsymbol{\sigma}|S,T})\le\mu} \langle Q, u_{\boldsymbol{\sigma}} \rangle \le \sqrt{\frac{m\gamma}{\eta - m\gamma}} \cdot \sup_h |u_{\boldsymbol{\sigma}}(h)| \le \sqrt{\frac{2m\gamma}{\eta}} \cdot m.$$

For any pair $(S, T)$ *not* satisfying (9), we can use the trivial bound

$$\sup_{\mathsf{D}(Q\|P_{S_T^\sigma})\le\mu} \langle Q, u_{\boldsymbol{\sigma}} \rangle - \sup_{\mathsf{D}(Q'\|\mathcal{P}'_{\boldsymbol{\sigma}|S,T})\le\mu} \langle Q, u_{\boldsymbol{\sigma}} \rangle \le m.$$

This allows us to bound the Rademacher complexity as follows:

$$\mathfrak{R}_m^\diamond(\mathcal{Q}_{m,\mu}) = \frac{1}{m} \mathop{\mathbb{E}}_{S,T,\boldsymbol{\sigma}}\left[\sup_{\mathsf{D}(Q\|P_{S_T^\sigma})\le\mu} \langle Q, u_{\boldsymbol{\sigma}} \rangle\right] \le \frac{1}{m} \mathop{\mathbb{E}}_{S,T,\boldsymbol{\sigma}}\left[\sup_{\mathsf{D}(Q\|\mathcal{P}'_{\boldsymbol{\sigma}|S,T})\le\mu} \langle Q, u_{\boldsymbol{\sigma}} \rangle\right] + \left(1 - \frac{1}{m}\right)\sqrt{\frac{2m\gamma}{\eta}} + \frac{1}{m}.$$

Now define $\Psi_{\boldsymbol{\sigma}|S,T}(Q)$ by $\Psi_{\boldsymbol{\sigma}|S,T}(Q) = \mathsf{D}(Q\|\mathcal{P}'_{\boldsymbol{\sigma}|S,T})$ if $\mathsf{D}(Q \parallel \mathcal{P}'_{\boldsymbol{\sigma}|S,T}) \le \mu$ and $+\infty$ otherwise. The conjugate function $\Psi_{\boldsymbol{\sigma}|S,T}^*(u) = \log\left(\mathbb{E}_{h\in\mathcal{P}'_{\boldsymbol{\sigma}|S,T}}[e^{u(h)}]\right)$, for all $u \in \mathbb{R}^{\mathcal{H}}$ [Mohri et al., 2018, Lemma B.37]. Continuing the bound on $\mathfrak{R}_m^\diamond(\mathcal{Q}_{m,\mu})$ above, for any $t > 0$,

$$\mathfrak{R}_m^\diamond(\mathcal{Q}_{m,\mu}) \le \frac{1}{mt} \mathop{\mathbb{E}}_{S,T,\boldsymbol{\sigma}}\left[\sup_{\mathsf{D}(Q\|\mathcal{P}'_{\boldsymbol{\sigma}|S,T})\le\mu} \langle Q, tu_{\boldsymbol{\sigma}} \rangle\right] + \sqrt{\frac{2m\gamma}{\eta}} + \frac{1}{m}$$

$$\le \frac{1}{mt} \mathop{\mathbb{E}}_{S,T,\boldsymbol{\sigma}}\left[\sup_{\Psi_{\boldsymbol{\sigma}|S,T}(Q)\le\mu} \Psi_{\boldsymbol{\sigma}|S,T}(Q) + \Psi_{\boldsymbol{\sigma}|S,T}^*(tu_{\boldsymbol{\sigma}})\right] + \sqrt{\frac{2m\gamma}{\eta}} + \frac{1}{m} \quad \text{(Fenchel inequality)}$$

$$\le \frac{\mu}{mt} + \frac{1}{mt} \mathop{\mathbb{E}}_{S,T,\boldsymbol{\sigma}}\left[\log\left(\mathop{\mathbb{E}}_{h\sim\mathcal{P}'_{\boldsymbol{\sigma}|S,T}}\left[e^{tu_{\boldsymbol{\sigma}}(h)}\right]\right)\right] + \sqrt{\frac{2m\gamma}{\eta}} + \frac{1}{m} \quad (\mu \text{ u.b., definition of } \Psi^*)$$

$$\leq \frac{\mu}{mt} + \frac{1}{mt}\Big[\log\Big(\mathbb{E}_{\boldsymbol{\sigma}}\mathbb{E}_{(S,T,h)\sim\mathcal{P}'_{\boldsymbol{\sigma}}}\big[e^{tu_{\boldsymbol{\sigma}}(h)}\big]\Big)\Big] + \sqrt{\frac{2m\gamma}{\eta}} + \frac{1}{m} \qquad \text{(Jensen's inequality)}$$

$$\leq \frac{\mu}{mt} + \frac{1}{mt}\Big[\log\Big(\mathbb{E}_{\boldsymbol{\sigma}}\mathbb{E}_{(S,T,h)\sim\mathcal{D}^{2m}\otimes\mathcal{P}}\big[e^{\kappa}e^{tu_{\boldsymbol{\sigma}}(h)}\big]\Big)\Big] + \sqrt{\frac{2m\gamma}{\eta}} + \frac{1}{m} \quad (\text{since } \mathsf{D}_{\infty}(\mathcal{P}'_{\boldsymbol{\sigma}}\|\mathcal{D}^{2m}\otimes\mathcal{P}) \leq \kappa)$$

$$\leq \frac{\mu}{mt} + \frac{1}{mt}\Big[\log\Big(\mathbb{E}_{(S,T,h)\sim\mathcal{D}^{2m}\otimes\mathcal{P}} e^{\frac{t^2 m}{2}}\Big)\Big] + \frac{\kappa}{mt} + \sqrt{\frac{2m\gamma}{\eta}} + \frac{1}{m} \qquad \text{(Hoeffding's lemma)}$$

$$= \frac{\mu}{mt} + \frac{t}{2} + \frac{\epsilon^2}{t} + \frac{\epsilon}{t}\sqrt{\frac{\log(2/\gamma)}{m}} + \sqrt{\frac{2m\gamma}{\eta}} + \frac{1}{m}.$$

Now we plug in $\gamma = \frac{\eta}{m^2}$ and choose $t = \sqrt{\frac{2\mu}{m} + 2\epsilon^2 + 2\epsilon\sqrt{\frac{\log(2m^2/\eta)}{m}}}$ to obtain the claimed bound.

$\square$

The requirement of the minimum probability $\eta > 0$ in Theorem 4 limits the applicability of the theorem to finite hypothesis sets $\mathcal{H}$. Via a similar proof technique, we can also derive the following PAC-Bayes bound in the case the priors have $\mathsf{D}_{\infty}$ stability without any requirement of a minimum probability. The precise bound and the proof appear in Appendix B.4.

**Theorem 5.** *Suppose the family of sample-dependent priors $(P_S)_{S\in\mathcal{Z}^m}$ has $\mathsf{D}_{\infty}$ sensitivity $\epsilon$. Then, for any $\delta > 0$, with probability at least $1 - \delta$ over the draw of the sample $S \sim \mathcal{D}^m$, the following inequality holds for all $Q \in \Delta(\mathcal{H})$: if $D = \max\{\mathsf{D}(Q \parallel P_S), 2\}$,*

$$\mathbb{E}_{\substack{h\sim Q \\ z\sim\mathcal{D}}}[L(h,z)] \leq \mathbb{E}_{h\sim Q}\Big[\frac{1}{m}\sum_{i=1}^{m}L(h,z_i)\Big] + O\Bigg(\sqrt{\frac{D}{m} + \epsilon^2 + \frac{\epsilon}{\sqrt{m}}} + \epsilon^{2/3}\mathfrak{R}_m(\mathcal{H})^{1/3} + \epsilon^{4/5}$$

$$+ \Big(\epsilon\mathfrak{R}_m(\mathcal{H}) + \epsilon\sqrt{\frac{\log(m^{1.5}D/\delta)}{m}} + \frac{1}{m}\Big)\sqrt{m\log\Big(\frac{D}{\delta}\Big)}\Bigg).$$

The above bound can also be extended to the case where the priors define an $(\epsilon,\delta)$-differentially private mechanism for some $\delta > 0$, instead of a pure $\epsilon$-differentially private mechanism, as required by Theorem 5. The precise bound in the theorem below and proof can be found in Appendix B.5.

**Theorem 6.** *Assume that $\epsilon \geq 0$ and $\delta \in [0, \frac{Ce^{-16m\epsilon}}{m^2}]$ for some constant $C$. Suppose the family of sample-dependent priors $(P_S)_{S\in\mathcal{Z}^m}$ satisfy the property that $\mathsf{D}_{\infty}^{\delta}(P_S\|P_{S'}) \leq \epsilon$ for all $S, S' \in \mathcal{Z}^m$ differing in exactly one point. Then, for any $\nu > 0$, with probability at least $1 - \nu$ over the draw of the sample $S \sim \mathcal{D}^m$, the following inequality holds for all $Q \in \Delta(\mathcal{H})$: if $D = \max\{\mathsf{D}(Q \parallel P_S), 2\}$,*

$$\mathbb{E}_{\substack{h\sim Q \\ z\sim\mathcal{D}}}[L(h,z)] \leq \mathbb{E}_{h\sim Q}\Big[\frac{1}{m}\sum_{i=1}^{m}L(h,z_i)\Big] + O\Bigg(\sqrt{\frac{D}{m} + \epsilon^2 + \frac{\epsilon}{\sqrt{m}}} + \epsilon^{2/3}\mathfrak{R}_m(\mathcal{H})^{1/3} + \epsilon^{4/5} + \frac{\sqrt{\delta}}{\epsilon^{3/2}}$$

$$+ \Big(\epsilon\mathfrak{R}_m(\mathcal{H}) + \epsilon\sqrt{\frac{\log(m^{1.5}D/\nu)}{m}} + \frac{1}{m}\Big)\sqrt{m\log\Big(\frac{D}{\nu}\Big)}\Bigg).$$

Similarly, one can consider studying other variants of the above theorems based on different sensitivity assumptions via the template above. A crucial component in implementing the template is effective control of the stability term $\beta$ in various settings. Below, we give a few lemmas which provide such control, some of which are used in the proof of Theorem 4. Proofs of these lemmas can be found in Appendices B.6, B.7 and B.8 respectively.

**Lemma 5.** *Suppose $\|P_S - P_{S'}\|_1 \leq \epsilon$ for all $S, S' \in \mathcal{Z}^m$ differing by exactly one point. For some $\mu \geq 0$, define the sample-dependent set of distributions as $\mathcal{Q}_{S,\mu} := \{Q\colon \mathsf{D}(Q \parallel P_S) \leq \mu\}$, and the corresponding family to be $\mathcal{Q}_{m,\mu} = (\mathcal{Q}_{S,\mu})_{S\in\mathcal{Z}^m}$. Then $\mathcal{Q}_{m,\mu}$ is $\beta$-stable for $\beta = \min\Big\{\frac{\epsilon d_{\infty}}{\sqrt{2\mu}}, \sqrt{\frac{\epsilon d_{\infty}}{2}}\Big\}$, where $d_{\infty} := \sup_{S,S',Q\in\mathcal{Q}_{S,\mu}}\big\|\frac{Q}{P_{S'}}\big\|_{\infty}$.*

**Lemma 6.** *Suppose $\mathsf{D}_{\infty}(P_S \parallel P_{S'}) \leq \epsilon$ for all $S, S' \in \mathcal{Z}^m$ differing by exactly one point. For some $\mu \geq 0$, define the sample-dependent set of distributions as $\mathcal{Q}_{S,\mu} := \{Q\colon \mathsf{D}(Q \parallel P_S) \leq \mu\}$, and the corresponding family to be $\mathcal{Q}_{m,\mu} = (\mathcal{Q}_{S,\mu})_{S\in\mathcal{Z}^m}$. Then $\mathcal{Q}_{m,\mu}$ is $\beta$-stable for $\beta = \min\Big\{2\epsilon, \frac{\epsilon}{\sqrt{2\mu}}, \sqrt{\frac{\epsilon}{2}}\Big\}$.*

**Lemma 7.** *Suppose $\|P_S - P_{S'}\|_1 \le \epsilon$ for all $S, S' \in \mathcal{Z}^m$ differing by exactly one point. For some $\mu \ge 0$, define the sample-dependent set of distributions as $\mathcal{Q}_{S,\mu} := \{Q : \|Q - P_S\|_1 \le \mu\}$, and the corresponding family to be $\mathcal{Q}_{m,\mu} = (\mathcal{Q}_{S,\mu})_{S \in \mathcal{Z}^m}$. Then $\mathcal{Q}_{m,\mu}$ is $\beta$-stable for $\beta = \frac{\epsilon}{2}$.*

We conclude with a brief discussion of some important considerations when applying these bounds.

**Choice of $\epsilon$.**  Depending on the choice of the function $S \mapsto P_S$ (for $S \in \mathcal{Z}^m$) and divergence function $\rho$, the family of priors $(P_S)_{S \in \mathcal{Z}^m}$ can be made to have varying sensitivity $\epsilon$. A larger value of $\epsilon$ provides greater freedom in selecting a sample-dependent prior, thus making it possible to cleverly choose a prior $P_S$ *closer* to the posterior $Q$ in order to decrease $\mathsf{D}(Q \parallel P_S)$. However, the price of such flexibility in our bounds is captured via additive terms that increase with $\epsilon$. In Corollary 1, we see that to derive $O\left(\frac{1}{\sqrt{m}}\right)$ rates, one must choose $\epsilon = O\left(\frac{1}{m}\right)$. In our refined bounds based on hypothesis set stability, a weaker $\epsilon = O\left(\frac{1}{\sqrt{m}}\right)$ suffices to obtain such $O\left(\frac{1}{\sqrt{m}}\right)$ rates, assuming that the Rademacher complexity of the base hypothesis class $\mathcal{H}$, $\mathfrak{R}_m(\mathcal{H})$, scales as $O\left(\frac{1}{\sqrt{m}}\right)$.

**Choice of $\mu$.**  Our bounds are intentionally presented as standard PAC-Bayes bounds - i.e., for all $Q \in \Delta(\mathcal{H})$ - eliminating the need for an explicit choice of $\mu$ to control the size of the sample-dependent sets of priors. However, if one were to fix $\mu$ (possibly as a function of $m$) a priori and then only consider those posteriors contained in the sample-dependent set $\mathcal{Q}_{S,\mu}$, then the application of Lemma 3 would be unnecessary.

**A general application recipe.**  A general strategy for obtaining an $\epsilon$-sensitive family of priors is to leverage any existing algorithm known to generate "parameter-stable" hypotheses (i.e., those for which the final parameters are close in some metric upon swapping a single element of the training set, under some parameterization of the hypothesis class). A notable example is the application of gradient descent with a limited number of parameter updates, under certain conditions discussed in [Feldman and Vondrak, 2018, 2019, Hardt et al., 2016]. Let $w_S = \mathcal{A}(S)$ denote the parameters found by running such a parameter-stable algorithm $\mathcal{A}$ on sample $S$. One natural prior $P_S$ is then a Gaussian distribution centered at $w_S$. Concretely, in a neural network setting, one could imagine running gradient descent on a sample $S$ for a limited number of iterations to obtain a parameter-stable prior $P_S$ and then continuing training on $S$ to generate a posterior $Q$. The art is in choosing an appropriate family of priors sufficiently close to the posteriors for the application in question; in the above neural network setting, this involves choosing the number of iterations to use in generating $P_S$, which highlights the tradeoff between $\epsilon$ and $\mathsf{D}(Q \parallel P_S)$.

## 5   Conclusion

We presented a general framework for deriving PAC-Bayesian learning bounds with sample-dependent priors, by leveraging the recently introduced notion of hypothesis set stability [Foster et al., 2019]. Our bounds include covering-number-based bounds, as well as bounds specifically tailored to priors satisfying a sensitivity condition, upon swapping an element of the sample.

This provides a broad framework for deriving PAC-Bayesian bounds that takes advantage of the full training sample when generating a prior. Much of our results can be further extended to the use of arbitrary Bregman divergences, instead of the specific (unnormalized) relative entropy. In particular, our Rademacher complexity analysis can be used similarly to derive upper bounds in that case.

A by-product of our analysis is a finer learning guarantee for sample-dependent hypothesis sets without any specific assumption about the closeness of these sample-dependent sets. This can provide a powerful tool for the analysis of broad collections of learning algorithms, or the design of new algorithms. While we leveraged these guarantees here, in particular by considering a closeness based on hypothesis set stability implied by the sensitivity of the priors, an important research direction is that of exploring alternative notions of closeness that can be significant for the theory of sample-dependent learning guarantees.

## Broader Impact

Due to the theoretical nature of this paper, we currently cannot foresee any short-to-medium-term negative societal impact. In general, we believe that the *short*-term societal impact is extremely limited. However, we hope that, in the medium-to-long term, such bounds - or other theoretical work that follows from such bounds - will play a role in furthering our understanding of the differences in generalization performance among various algorithms. We believe that such understanding is very important when deploying models in real-world settings and when attempting to design new algorithms that generalize even better. PAC-Bayes bounds, in particular, have shown some promise in explaining the generalization performance of neural networks, which are widely used in practice. Thus, beyond a general claim about various machine learning algorithms, we think it is possible that our bounds or those inspired by them can contribute to the community's understanding of (and expectations for) neural networks. Due to the widespread use of neural networks, such improved understanding can have a significant positive impact.

## Funding Disclosure & Acknowledgments

We have no funding to disclose. We thank Vitaly Feldman for pointers relevant to the proof of Theorem 6.

## Footnotes

[1]The same notion has also been called *data-dependent* priors in the literature [Dziugaite and Roy, 2018a, Negrea et al., 2019, Dziugaite et al., 2020, Haghifam et al., 2020].

[2]Extension to continuous domains is straightforward using standard measure-theoretic formulations.

[3]Technically, this is the Rademacher complexity of the class $\mathcal{G} = \{z \mapsto L(h, z) \colon h \in \mathcal{H}\}$, however we define it in this way by absorbing the loss function for clarity of notation.

[4]Theorem 20 in [Dwork et al., 2015] is stated in terms of approximate max-information; here we state the equivalent bound in terms of approximate infinite-Rényi divergence: see Definition 10 in [Dwork et al., 2015].

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
