[Supplementary Material]

# A Proofs of results in Section 3

## A.1 Proof of Theorem 1

Here, we present the full proof of Theorem 1, with the precise bound spelled out. To present the theorem, recall the definition of $\tilde{\mathfrak{R}}_{U,m}(\mathcal{H})$ in (5): let $m, n$ be two positive integers, and let $U = (z_1, z_2, \ldots, z_{m+n}) \in \mathcal{Z}^{m+n}$ be a sample set. Then we define a notion of Rademacher complexity $\tilde{\mathfrak{R}}_{U,m}(\mathcal{H})$ as follows: if $\boldsymbol{\sigma}$ is a vector of $(m + n)$ independent random variables taking value $\frac{m+n}{n}$ with probability $\frac{n}{m+n}$ and value $-\frac{m+n}{m}$ with probability $\frac{m}{m+n}$, then

$$\tilde{\mathfrak{R}}_{U,m}(\mathcal{H}) := \frac{1}{m+n} \mathop{\mathbb{E}}_{\boldsymbol{\sigma}} \left[ \sup_{h \in \mathcal{H}} \left| \sum_{i=1}^{m+n} \sigma_i L(h, z_i) \right| \right]$$

Furthermore, define $\tilde{\mathfrak{R}}_{m,n} = \mathbb{E}_U[\tilde{\mathfrak{R}}_{U,m}(\mathcal{H})]$.

The bound of Theorem 1 as stated in Section 3 is for the special case $m = n$, and is stated in terms of the standard Rademacher complexity $\mathfrak{R}_{2m}(\mathcal{H})$. This follows from the following bound:

**Lemma 8.** *If $m = n$, then $\tilde{\mathfrak{R}}_{U,m}(\mathcal{H}) \leq 4\mathfrak{R}_U(\mathcal{H})$.*

*Proof.* Since $m = n$, $\boldsymbol{\sigma}$ is a vector of $2m$ variables taking values in $\{-2, 2\}$ uniformly at random.

$$\begin{aligned}
\tilde{\mathfrak{R}}_{U,m}(\mathcal{H}) &= \frac{1}{2m} \mathop{\mathbb{E}}_{\boldsymbol{\sigma}} \left[ \sup_{h \in \mathcal{H}} \left| \sum_{i=1}^{2m} \sigma_i L(h, z_i) \right| \right] \\
&= \frac{1}{2m} \mathop{\mathbb{E}}_{\boldsymbol{\sigma}} \left[ \sup_{\substack{h \in \mathcal{H} \\ s \in \{-1,+1\}}} s \sum_{i=1}^{2m} \sigma_i L(h, z_i) \right] \\
&\leq \frac{1}{2m} \mathop{\mathbb{E}}_{\boldsymbol{\sigma}} \left[ \sup_{h \in \mathcal{H}} \sum_{i=1}^{2m} \sigma_i L(h, z_i) \right] + \frac{1}{2m} \mathop{\mathbb{E}}_{\boldsymbol{\sigma}} \left[ \sup_{h \in \mathcal{H}} \sum_{i=1}^{2m} -\sigma_i L(h, z_i) \right] \\
&= 4\mathfrak{R}_U(\mathcal{H}).
\end{aligned}$$

$\square$

**Theorem 1.** *Let $P_S \in \Delta(\mathcal{H})$ be a prior over $\mathcal{H}$ determined by the choice of $S \in \mathcal{Z}^m$, and let $n$ be a positive integer. Then, for any $\delta > 0$, with probability at least $1 - \delta$ over the draw of the sample $S \sim \mathcal{D}^m$, the following inequality holds for all $Q \in \Delta(\mathcal{H})$, if $D := \max\{\mathsf{D}(Q\|P_S), 2\}$,*

$$\mathop{\mathbb{E}}_{\substack{h \sim Q \\ z \sim \mathcal{D}}} [L(h, z)] \leq \mathop{\mathbb{E}}_{\substack{h \sim Q \\ z \sim S}} [L(h, z)] + \inf_{\alpha \geq 0} \sqrt{2 \left( 2D + \alpha + \log \mathcal{N}(\alpha, m, n, \mathsf{D}_\infty) \right) \left( \frac{1}{m} + \frac{1}{n} \right)^3 mn} \tag{10}$$
$$+ 3\sqrt{\left( \frac{1}{m} + \frac{1}{n} \right) \log\left( \frac{4D}{\delta} \right)} + 2\sqrt{\left( \frac{1}{m} + \frac{1}{n} \right)^3 mn \log\left( \frac{8eD}{\delta} \right)}.$$

*Similarly, for any $\delta > 0$, with probability at least $1 - \delta$ over the draw of the sample $S \sim \mathcal{D}^m$, the following inequality holds for all $Q \in \Delta(\mathcal{H})$:*

$$\mathop{\mathbb{E}}_{\substack{h \sim Q \\ z \sim \mathcal{D}}} [L(h, z)] \leq \mathop{\mathbb{E}}_{\substack{h \sim Q \\ z \sim S}} [L(h, z)] + \inf_{\alpha \geq 0} 2(2\sqrt{D} + \alpha)\tilde{\mathfrak{R}}_{m,n}(\mathcal{H}) + \sqrt{2 \log(\mathcal{N}(\alpha, m, n, \ell_1)) \left( \frac{1}{m} + \frac{1}{n} \right)^3 mn}$$
$$+ 3\sqrt{\left( \frac{1}{m} + \frac{1}{n} \right) \log\left( \frac{4D}{\delta} \right)} + 2\sqrt{\left( \frac{1}{m} + \frac{1}{n} \right)^3 mn \log\left( \frac{8eD}{\delta} \right)}. \tag{11}$$

*Proof.* Fix $\mu > 0$ and define the sample-dependent hypothesis set as

$$\mathcal{Q}_{S,\mu} = \left\{ Q \in \Delta(\mathcal{H}) \colon \mathsf{D}(Q\|P_S) \leq \mu \right\},$$

where $\Delta(\mathcal{H})$ is the family of all distributions defined over $\mathcal{H}$. We define the loss of $Q \in \Delta(\mathcal{H})$ over the labeled sample $z = (x, y) \in \mathcal{Z}$ as $\ell(Q, z) = \langle Q, L_z \rangle$. Thus, the expected loss of $Q$ is

$$\mathop{\mathbb{E}}_{z \sim \mathcal{D}} [\ell(Q, z)] = \mathop{\mathbb{E}}_{\substack{h \sim Q \\ z \sim \mathcal{D}}} [L(h, z)].$$

We also define the sample-indexed family of sample-dependent hypothesis sets $\mathcal{Q}_{m,\mu} = (\mathcal{Q}_{S,\mu})_{S \in \mathcal{Z}^m}$ and the $U$-restricted union of sample-dependent hypothesis sets $\overline{\mathcal{Q}}_{U,m,\mu} = \bigcup_{\substack{S \in \mathcal{Z}^m \\ S \subseteq U}} \mathcal{Q}_{S,\mu}$.

In view of that, by Theorem 2, for any $\delta > 0$, with probability $1 - \delta$ over the draw of a sample $S \sim \mathcal{D}^m$, the following holds for any $Q \in \mathcal{H}_{S,\mu}$:

$$\mathop{\mathbb{E}}_{\substack{h \sim Q \\ z \sim \mathcal{D}}} \big[ L(h,z) \big] \le \mathop{\mathbb{E}}_{\substack{h \sim Q \\ z \sim S}} \big[ L(h,z) \big] + 2 \max_{U \in \mathcal{Z}^{m+n}} \widehat{\mathfrak{R}}^{\diamond}_{U,m}(\mathcal{Q}_{m,\mu}) + 3\sqrt{\left(\tfrac{1}{m} + \tfrac{1}{n}\right)\log\left(\tfrac{2}{\delta}\right)} + 2\sqrt{\left(\tfrac{1}{m} + \tfrac{1}{n}\right)^3 mn},$$

where $\widehat{\mathfrak{R}}^{\diamond}_{U,m}(\mathcal{Q}_{m,\mu})$ is defined for any $U = (z_1, \ldots, z_{m+n}) \in \mathcal{Z}^{m+n}$ as follows: if $\boldsymbol{\sigma}$ is a vector of $(m+n)$ independent random variables taking value $\frac{m+n}{n}$ with probability $\frac{n}{m+n}$ and value $-\frac{m+n}{m}$ with probability $\frac{m}{m+n}$, then

$$\widehat{\mathfrak{R}}^{\diamond}_{U,m}(\mathcal{Q}_{m,\mu}) = \mathop{\mathbb{E}}_{\boldsymbol{\sigma}} \left[ \sup_{Q \in \overline{\mathcal{Q}}_{U,m,\mu}} \frac{1}{m+n} \sum_{i=1}^{m+n} \sigma_i \langle Q, L_{z_i} \rangle \right].$$

Via covering number arguments for $\mathsf{D}_\infty$ (Lemma 1) and $\ell_1$ (Lemma 2) we derive bounds on $\widehat{\mathfrak{R}}^{\diamond}_{U,m}(\mathcal{Q}_{m,\mu})$. The bounds in the theorem then follow by applying Lemma 3. $\qquad\square$

## A.2 Proof of Lemma 1

**Lemma 1.** *For any $\alpha \ge 0$, we have*

$$\widehat{\mathfrak{R}}^{\diamond}_{U,m}(\mathcal{Q}_{m,\mu}) \le \sqrt{\left(\frac{\mu + \alpha + \log\mathcal{N}(\alpha, U, \mathsf{D}_\infty)}{2}\right)\left(\frac{1}{m} + \frac{1}{n}\right)^3 mn}.$$

*Proof.* Let $C$ be a covering for $U$ under $\mathsf{D}_\infty$ at scale $\alpha$ of size $\mathcal{N}(\alpha, U, \mathsf{D}_\infty)$. Define $\mathcal{G}_{U,m,\mu+\alpha}$ as

$$\mathcal{G}_{U,m,\mu+\alpha} := \{Q \in \Delta(\mathcal{H}) : \exists P \in C \text{ s.t. } \mathsf{D}(Q\|P) \le \mu + \alpha\}.$$

Now, let $Q \in \overline{\mathcal{H}}_{U,m,\mu}$. Then there exists a some subset $S$ of $U$ of size $m$, such that $\mathsf{D}(Q\|P_S) \le \mu$. Since $C$ is a covering for $U$ under $\mathsf{D}_\infty$ at scale $\alpha$, there exists a distribution $P' \in C$ such that $\mathsf{D}_\infty(P\|P') \le \alpha$. We have $\mathsf{D}(Q\|P') \le \mathsf{D}(Q\|P) + \mathsf{D}_\infty(P\|P') \le \mu + \alpha$. Thus, $Q \in \mathcal{G}_{U,m,\mu+\alpha}$. This implies that $\overline{\mathcal{H}}_{U,m,\mu} \subseteq \mathcal{G}_{U,m,\mu+\alpha}$.

In the following derivation, we will use the shorthand $u_{\boldsymbol{\sigma}}(h) = \sum_{i=1}^{m+n} \sigma_i L(h, z_i)$, so that $\sum_{i=1}^{m+n} \sigma_i \langle Q, L_{z_i} \rangle = \langle Q, u_{\boldsymbol{\sigma}} \rangle$. For any $P \in C$ and $Q \in \Delta(\mathcal{H})$, define $\Psi_P(Q)$ by $\Psi_S(Q) = \mathsf{D}(Q\|P_S)$ if $\mathsf{D}(Q\|P_S) \le \mu + \alpha$ and $+\infty$ otherwise. It is known that the conjugate function $\Psi_P^*$ of $\Psi_P$ is given by $\Psi_P^*(u) = \log\big(\mathbb{E}_{h \in P}[e^{u(h)}]\big)$, for all $u \in \mathbb{R}^{\mathcal{H}}$ (see for example [Mohri et al., 2018, Lemma B.37]). We now upper bound the transductive Rademacher complexity term as follows:

$$\widehat{\mathfrak{R}}^{\diamond}_{U,m}(\mathcal{Q}_{m,\mu}) = \frac{1}{m+n} \mathop{\mathbb{E}}_{\boldsymbol{\sigma}} \left[ \sup_{Q \in \overline{\mathcal{H}}_{U,m,\mu}} \langle Q, u_{\boldsymbol{\sigma}} \rangle \right] \qquad\qquad \text{(definition of } u_{\boldsymbol{\sigma}})$$

$$\le \frac{1}{m+n} \mathop{\mathbb{E}}_{\boldsymbol{\sigma}} \left[ \sup_{Q \in \mathcal{G}_{U,m,\mu+\alpha}} \langle Q, u_{\boldsymbol{\sigma}} \rangle \right] \qquad\qquad (\overline{\mathcal{H}}_{U,m,\mu} \subseteq \mathcal{G}_{U,m,\mu+\alpha})$$

$$= \frac{1}{(m+n)t} \mathop{\mathbb{E}}_{\boldsymbol{\sigma}} \left[ \sup_{Q \in \mathcal{G}_{U,m,\mu+\alpha}} \langle Q, t u_{\boldsymbol{\sigma}} \rangle \right] \qquad\qquad (t > 0)$$

$$= \frac{1}{(m+n)t} \mathop{\mathbb{E}}_{\boldsymbol{\sigma}} \left[ \sup_{P \in C} \sup_{Q:\, \mathsf{D}(Q\|P) \le \mu+\alpha} \langle Q, t u_{\boldsymbol{\sigma}} \rangle \right] \qquad\qquad \text{(iterated sup)}$$

$$\le \frac{1}{(m+n)t} \mathop{\mathbb{E}}_{\boldsymbol{\sigma}} \left[ \sup_{P \in C} \sup_{Q:\, \mathsf{D}(Q\|P) \le \mu+\alpha} \big[\Psi_P(Q) + \Psi_P^*(t u_{\boldsymbol{\sigma}})\big] \right] \qquad \text{(Fenchel inequality)}$$

$$\leq \frac{1}{(m+n)t} \mathbb{E}_{\boldsymbol{\sigma}} \left[ \sup_{P \in C} \left[ \mu + \alpha + \Psi_S^*(tu_{\boldsymbol{\sigma}}) \right] \right] \qquad \text{(definition of } \Psi_P(Q))$$

$$= \frac{\mu + \alpha}{(m+n)t} + \frac{1}{(m+n)t} \mathbb{E}_{\boldsymbol{\sigma}} \left[ \sup_{P \in C} \Psi_P^*(tu_{\boldsymbol{\sigma}}) \right] \qquad \text{(distribute)}$$

$$= \frac{\mu + \alpha}{(m+n)t} + \frac{1}{(m+n)t} \mathbb{E}_{\boldsymbol{\sigma}} \left[ \sup_{P \in C} \log \left( \mathbb{E}_{h \sim P} [e^{tu_{\boldsymbol{\sigma}}(h)}] \right) \right] \qquad \text{(definition of } \Psi_P^*)$$

We now upper bound $\mathbb{E}_{\boldsymbol{\sigma}} \left[ \sup_{P \in C} \log \left( \mathbb{E}_{h \sim P}[e^{tu_{\boldsymbol{\sigma}}(h)}] \right) \right]$ as follows:

$$\mathbb{E}_{\boldsymbol{\sigma}} \left[ \sup_{P \in C} \log \left( \mathbb{E}_{h \sim P}[e^{tu_{\boldsymbol{\sigma}}(h)}] \right) \right] = \mathbb{E}_{\boldsymbol{\sigma}} \left[ \log \left( \sup_{P \in C} \mathbb{E}_{h \sim P}[e^{tu_{\boldsymbol{\sigma}}(h)}] \right) \right] \qquad \text{(log is mon. incr.)}$$

$$\leq \log \left[ \mathbb{E}_{\boldsymbol{\sigma}} \left( \sup_{P \in C} \mathbb{E}_{h \sim P}[e^{tu_{\boldsymbol{\sigma}}(h)}] \right) \right] \qquad \text{(Jensen's inequality)}$$

$$\leq \log \left[ \mathbb{E}_{\boldsymbol{\sigma}} \left( \sum_{P \in C} \mathbb{E}_{h \sim P}[e^{tu_{\boldsymbol{\sigma}}(h)}] \right) \right] \qquad \text{(nonnegative terms)}$$

$$= \log \left[ \sum_{P \in C} \mathbb{E}_{h \sim P} \mathbb{E}_{\boldsymbol{\sigma}}[e^{tu_{\boldsymbol{\sigma}}(h)}] \right] \qquad \text{(lin. of expectation; } h, \boldsymbol{\sigma} \text{ indep.)}$$

$$= \log \left[ \sum_{P \in C} \mathbb{E}_{h \sim P} \mathbb{E}_{\boldsymbol{\sigma}} \left[ e^{t \sum_{i=1}^{m+n} \sigma_i L(h, z_i^U)} \right] \right] \qquad \text{(def. of } u_{\boldsymbol{\sigma}}(h))$$

$$= \log \left[ \sum_{P \in C} \mathbb{E}_{h \sim P} \left[ \prod_{i=1}^{m+n} \mathbb{E}_{\sigma_i} e^{t \sigma_i L(h, z_i^U)} \right] \right] \qquad \text{(indep. entries of } \boldsymbol{\sigma})$$

$$\leq \log \left[ \sum_{P \in C} \mathbb{E}_{h \sim P} \left[ e^{\frac{t^2 (m+n)^5}{8(mn)^2}} \right] \right] \qquad \text{(Hoeffding's lemma)}$$

$$= \log \left[ \sum_{P \in C} e^{\frac{t^2 (m+n)^5}{8(mn)^2}} \right] \qquad \text{(no dep. on } h)$$

$$= \log \left[ |C| \cdot e^{\frac{t^2 (m+n)^5}{8(mn)^2}} \right] \qquad \text{(all terms equal)}$$

$$= \log |C| + \frac{t^2 (m+n)^5}{8(mn)^2}.$$

Plugging this back in, we get:

$$\widehat{\mathfrak{R}}_{U,m}^{\diamond}(\mathcal{Q}_{m,\mu}) \leq \frac{\mu + \alpha}{(m+n)t} + \frac{1}{(m+n)t} \left[ \log |C| + \frac{t^2 (m+n)^5}{8(mn)^2} \right]$$

$$= \frac{\mu + \alpha + \log |C|}{(m+n)t} + \frac{t(m+n)^4}{8(mn)^2}.$$

We find that $t = \sqrt{\frac{8(mn)^2(\mu+\alpha+\log|C|)}{(m+n)^5}}$ minimizes the bound.

Plugging this optimal $t$ back in, we obtain:

$$\widehat{\mathfrak{R}}_{U,m}^{\diamond}(\mathcal{Q}_{m,\mu}) \leq \sqrt{\frac{(\mu + \alpha + \log |C|)(m+n)^3}{2(mn)^2}} = \sqrt{\left( \frac{\mu + \alpha + \log |C|}{2} \right) \left( \frac{1}{m} + \frac{1}{n} \right)^3 mn}.$$

$\square$

### A.3   Proof of Lemma 2

**Lemma 2.** *For any $\alpha \geq 0$, we have*

$$\widehat{\mathfrak{R}}_{U,m}^{\diamond}(\mathcal{Q}_{m,\mu}) \leq (\sqrt{2\mu} + \alpha) \tilde{\mathfrak{R}}_{U,m}(\mathcal{H}) + \sqrt{\frac{\log \mathcal{N}(\alpha, U, \ell_1)}{2} \left( \frac{1}{m} + \frac{1}{n} \right)^3 mn}.$$

*Proof.* Let $C$ be a covering for $U$ under $\ell_1$ at scale $\alpha$ of size $\mathcal{N}(\alpha, U, \ell_1)$. Let $\mathcal{G}_{U,m,\sqrt{2\mu}+\alpha}$ be the union of all the $\ell_1$ balls of radius $\sqrt{2\mu} + \alpha$ around distributions in $C$, i.e.

$$\mathcal{G}_{U,m,\sqrt{2\mu}} = \{Q \in \Delta(\mathcal{H}) : \exists P \in C \text{ s.t. } \|Q - P\|_1 \leq \sqrt{2\mu} + \alpha\}.$$

Now, let $Q \in \overline{\mathcal{H}}_{U,m,\mu}$. By Pinsker's inequality, for some subset $S$ of $U$ of size $m$, we have $\|Q - P_S\|_1 \leq \sqrt{2\mu}$. Since $C$ is a covering for $U$ under $\ell_1$ at scale $\alpha$, there exists a distribution $P \in C$ such that $\|P_S - P\|_1 \leq \alpha$. This implies that $\|Q - P\|_1 \leq \sqrt{2\mu} + \alpha$, so $Q \in \mathcal{G}_{U,m,\sqrt{2\mu}+\alpha}$. Hence $\overline{\mathcal{H}}_{U,m,\mu} \subseteq \mathcal{G}_{U,m,\sqrt{2\mu}+\alpha}$. In the following derivation, we will use the shorthand $u_{\boldsymbol{\sigma}}(h) = \sum_{i=1}^{m+n} \sigma_i L(h, z_i)$, so that $\sum_{i=1}^{m+n} \sigma_i \langle Q, L_{z_i} \rangle = \langle Q, u_{\boldsymbol{\sigma}} \rangle$. We can now proceed the bound the Rademacher complexity as follows:

$$\widehat{\mathfrak{R}}^{\diamond}_{U,m}(\mathcal{Q}_{m,\mu}) = \frac{1}{m+n} \mathop{\mathbb{E}}_{\boldsymbol{\sigma}} \left[ \sup_{Q \in \overline{\mathcal{H}}_{U,m,\mu}} \langle Q, u_{\boldsymbol{\sigma}} \rangle \right]$$

$$\leq \frac{1}{m+n} \mathop{\mathbb{E}}_{\boldsymbol{\sigma}} \left[ \sup_{Q \in \mathcal{G}_{U,m,\sqrt{2\mu}+\alpha}} \langle Q, u_{\boldsymbol{\sigma}} \rangle \right]$$

$$\leq \frac{1}{m+n} \mathop{\mathbb{E}}_{\boldsymbol{\sigma}} \left[ \sup_{P \in C} \langle P, u_{\boldsymbol{\sigma}} \rangle \right] + (\sqrt{2\mu} + \alpha) \tilde{\mathfrak{R}}_{U,m}(\mathcal{H}).$$

The last inequality follows since for any $Q \in \mathcal{G}_{U,m,\sqrt{2\mu}+\alpha}$ there exists a distribution $P \in C$ such that $\|Q - P\|_1 \leq \sqrt{2\mu} + \alpha$, and so we have

$$\mathop{\mathbb{E}}_{\boldsymbol{\sigma}}[|\langle Q - P, u_{\boldsymbol{\sigma}} \rangle|] \leq \mathop{\mathbb{E}}_{\boldsymbol{\sigma}}[\|Q - P\|_1 \|u_{\boldsymbol{\sigma}}\|_\infty] \leq (\sqrt{2\mu} + \alpha) \mathop{\mathbb{E}}_{\boldsymbol{\sigma}}[\|u_{\boldsymbol{\sigma}}\|_\infty] = (\sqrt{2\mu} + \alpha)(m+n) \tilde{\mathfrak{R}}_{U,m}(\mathcal{H}).$$

Now, define $v : \Delta(\mathcal{H}) \to [0,1]^{m+n}$ as $v(P)_i = \mathbb{E}_{h \sim P}[L(h, z_i)]$. Note that $\langle P, u_{\boldsymbol{\sigma}} \rangle = \langle \boldsymbol{\sigma}, v(P) \rangle$, and so

$$\mathop{\mathbb{E}}_{\boldsymbol{\sigma}} \left[ \sup_{P \in C} \langle P, u_{\boldsymbol{\sigma}} \rangle \right] = \mathop{\mathbb{E}}_{\boldsymbol{\sigma}} \left[ \sup_{P \in C} \langle \boldsymbol{\sigma}, v(P) \rangle \right].$$

We can now bound $\mathbb{E}_{\boldsymbol{\sigma}} \left[ \sup_{P \in C} \langle \boldsymbol{\sigma}, v(P) \rangle \right]$ by a version of Massart's lemma which applies to non-Rademacher (but still zero mean) random variables $\boldsymbol{\sigma}$, as follows: let $t > 0$ to be chosen momentarily. We have

$$\exp \left( t \mathop{\mathbb{E}}_{\boldsymbol{\sigma}} \left[ \sup_{P \in C} \langle \boldsymbol{\sigma}, v(P) \rangle \right] \right) \leq \mathop{\mathbb{E}}_{\boldsymbol{\sigma}} \left[ \exp \left( t \sup_{P \in C} \langle \boldsymbol{\sigma}, v(P) \rangle \right) \right] \qquad \text{(Jensen's inequality)}$$

$$\leq \mathop{\mathbb{E}}_{\boldsymbol{\sigma}} \left[ \sum_{P \in C} \exp \left( \langle \boldsymbol{\sigma}, t v(P) \rangle \right) \right]$$

$$= \mathop{\mathbb{E}}_{\boldsymbol{\sigma}} \left[ \sum_{P \in C} \prod_{i=1}^{m} \exp(t v(P)_i \sigma_i) \right]$$

$$= \sum_{P \in C} \prod_{i=1}^{m+n} \mathop{\mathbb{E}}_{\sigma_i} \left[ \exp(t v(P)_i \sigma_i) \right]$$

$$\leq |C| \exp \left( \frac{t^2 (m+n)^5}{8(mn)^2} \right) \qquad \text{(Hoeffding's lemma)}.$$

Thus,

$$\widehat{\mathfrak{R}}^{\diamond}_{U,m}(\mathcal{Q}_{m,\mu}) \leq \frac{1}{m+n} \mathop{\mathbb{E}}_{\boldsymbol{\sigma}} \left[ \sup_{P \in C} \langle \boldsymbol{\sigma}, v(P) \rangle \right] + (\sqrt{2\mu} + \alpha) \tilde{\mathfrak{R}}_{U,m}(\mathcal{H})$$

$$\leq \frac{\log |C|}{t(m+n)} + \frac{t(m+n)^4}{8(mn)^2} + 2(\sqrt{2\mu} + \alpha) \tilde{R}_{U,m}(\mathcal{H}).$$

Setting $t = \sqrt{\frac{8(mn)^2 (\log |C|)}{(m+n)^5}}$ to minimize the bound, we obtain:

$$\widehat{\mathfrak{R}}^{\diamond}_{U,m}(\mathcal{Q}_{m,\mu}) \leq \sqrt{\frac{(m+n)^3 \log |C|}{2(mn)^2}} + (\sqrt{2\mu} + \alpha) \tilde{\mathfrak{R}}_{U,m}(\mathcal{H}).$$

$\square$

## A.4 Proof of Lemma 3

**Lemma 3.** *Suppose the following bound holds with probability at least $1 - \delta$ over the choice of $S$: for all $Q \in \mathcal{Q}_{S,\mu}$,*

$$\mathop{\mathbb{E}}_{\substack{h \sim Q \\ z \sim \mathcal{D}}}\big[L(h,z)\big] \leq \mathop{\mathbb{E}}_{\substack{h \sim Q \\ z \sim S}}\big[L(h,z)\big] + f(\mu) + g(\delta),$$

*where $f$ is an increasing function of $\mu$ and $g$ is a decreasing function of $\delta$. Then, the following holds with probability at least $1 - \delta$ for all $Q \in \Delta(\mathcal{H})$:*

$$\mathop{\mathbb{E}}_{\substack{h \sim Q \\ z \sim \mathcal{D}}}\big[L(h,z)\big] \leq \mathop{\mathbb{E}}_{\substack{h \sim Q \\ z \sim S}}\big[L(h,z)\big] + f(2\max\{D(Q\|P_S),2\}) + g\left(\frac{\delta}{\max\{D(Q\|P_S),2\}}\right).$$

*Proof.* The proof follows [Kakade et al., 2008][Corollary 8]. First, define the sequences $(\mu_j)_{j=0}^{\infty}$ and $(\delta_j)_{j=0}^{\infty}$. Let $a = 4$, $\mu_j := a2^j$ and $\delta_j := 2^{-(j+1)}\delta$, so that $\sum_{j=0}^{\infty} \delta_j = \delta$.

By the union bound, we thus have that with probability at least $1 - \delta$ over the draw of a sample $S \sim \mathcal{D}^m$, for all $Q \in \Delta(\mathcal{H})$:

$$\mathop{\mathbb{E}}_{\substack{h \sim Q \\ z \sim \mathcal{D}}}\big[L(h,z)\big] \leq \mathop{\mathbb{E}}_{\substack{h \sim Q \\ z \sim S}}\big[L(h,z)\big] + f(\mu_j) + g(\delta_j) \tag{12}$$

where $\mu_j$ is the smallest element of $(\mu_j)_{j=0}^{\infty}$ such that $D(Q\|P_S) \leq \mu_j$ (i.e., since we have a sequence of bounds holding for increasing values of $\mu_j$, we choose the tightest applicable bound for each $Q$).

We now plug in the values of $\mu_j, \delta_j$:

$$\mathop{\mathbb{E}}_{\substack{h \sim Q \\ z \sim \mathcal{D}}}\big[L(h,z)\big] \leq \mathop{\mathbb{E}}_{\substack{h \sim Q \\ z \sim S}}\big[L(h,z)\big] + f(a2^j) + g(2^{-(j+1)}\delta) \tag{13}$$

and try to upper bound the RHS in terms of $D(Q\|P_S)$, eliminating any appearances of $j$ (i.e., we want a single bound that captures the sequence of bounds).

**Upper bound $\mu_j$:** By the assumption that $\mu_j$ is the smallest element of $(\mu_j)_{j=0}^{\infty}$ such that $D(Q\|P_S) \leq \mu_j$, we necessarily have $D(Q\|P_S) > \mu_{j-1}$ for $j \geq 1$. (For $j = 0$, this simply yields $D(Q\|P_S) \geq 0$, which will not help, so we need to handle $j = 0$ separately.)

For $j \geq 1$, we thus have $D(Q\|P_S) > \mu_{j-1} = a2^{j-1}$, so $2D(Q\|P_S) > a2^j$.

For $j = 0$, $a2^j = a$.

This yields:

$$a2^j \leq \max\{2D(Q\|P_S),a\} = 2\max\{D(Q\|P_S),2\}.$$

**Lower bound $\delta_j$:** Since $\delta_j = 2^{-(j+1)}\delta$, we use the same assumption as above to obtain $4D(Q\|P_S) > a2^{j+1}$ and then use the definition of $\delta_j$ to obtain the lower bound: $\delta_j > \frac{a\delta}{4D(Q\|P_S)}$ for $j \geq 1$. For $j = 0$, we simply have $\delta_j = \delta/2$ by definition. This yields:

$$\delta_j \geq \min\left\{\frac{a\delta}{4D(Q\|P_S)}, \delta/2\right\} = \frac{\delta}{\max\{D(Q\|P_S),2\}}.$$

The stated bound follows from the monotonicities of $f$ and $g$. $\qquad\square$

# B    Proofs of results in Section 4

## B.1    Proof of Theorem 3

We prove Theorem 3, with the exact bound explicitly spelled out:

**Theorem 3.** *Suppose $\mathcal{Q}_m = (\mathcal{Q}_S)_{S \in \mathcal{Z}^m}$ is $\beta$-uniformly stable. Then, for any $\delta > 0$, with probably at least $1 - \delta$ over the draw of the sample $S \sim \mathcal{D}^m$, the following holds for all $Q \in \mathcal{Q}_S$:*

$$
\mathop{\mathbb{E}}_{\substack{h \sim Q \\ z \sim \mathcal{D}}} \left[ L(h, z) \right] \leq \mathop{\mathbb{E}}_{h \sim Q} \left[ \frac{1}{m} \sum_{i=1}^{m} L(h, z_i) \right]
$$

$$
+ 2 \mathfrak{R}_m^{\diamond}(\mathcal{Q}_m) + \left( 2\beta \left( 2 \mathfrak{R}_m(\mathcal{H}) + \sqrt{\frac{\log(4m^{1.5}/\delta)}{2m}} \right) + \frac{1}{m} \right) \sqrt{8m \log\left( \frac{4}{\delta} \right)}.
$$

*Proof.* The proof is along the lines of the proof of Theorem 2 in [Foster et al., 2019] with a tighter analysis coming from the special structure in our setting. Specifically, for two samples $S, S' \in \mathcal{Z}^m$, define the function $\Psi(S, S')$ as follows:

$$
\Psi(S, S') = \sup_{Q \in \mathcal{Q}_S} \langle Q, \ell \rangle - \langle Q, \hat{\ell}_{S'} \rangle,
$$

where $\ell, \hat{\ell}_{S'} \in \mathfrak{R}^{\mathcal{H}}$ defined as $\ell(h) = \mathbb{E}_{z \sim \mathcal{D}}[L(h, z)]$ and $\hat{\ell}_{S'}(h) = \mathbb{E}_{z \sim S'}[L(h, z)]$, where $z \sim S'$ indicates uniform sampling from $S'$. The proof of the bound consists of applying McDiarmid's inequality to $\Psi(S, S)$. To do this, we need to analyze the sensitivity of this function, i.e. compute a bound on $|\Psi(S, S) - \Psi(S', S')|$ where $S'$ is a sample differing from $S$ in exactly one point. As in [Foster et al., 2019], we first observe that $\Psi(S, S) - \Psi(S, S') \leq \frac{1}{m}$, so now we turn to

$$
\Psi(S, S') - \Psi(S', S') = \sup_{Q \in \mathcal{Q}_S} \langle Q, \ell \rangle - \langle Q, \hat{\ell}_{S'} \rangle - \sup_{Q \in \mathcal{Q}_{S'}} \langle Q, \ell \rangle - \langle Q, \hat{\ell}_{S'} \rangle.
$$

By definition of the supremum, for any $\epsilon > 0$ there exists a $Q_\epsilon \in \mathcal{Q}_S$ such that

$$
\sup_{Q \in \mathcal{Q}_S} \langle Q, \ell \rangle - \langle Q, \hat{\ell}_{S'} \rangle - \epsilon \leq \sup_{Q \in \mathcal{Q}_S} \langle Q_\epsilon, \ell \rangle - \langle Q_\epsilon, \hat{\ell}_{S'} \rangle.
$$

Using the $\beta$-stability of $\mathcal{Q}_m = (\mathcal{Q}_S)_{S \in \mathcal{Z}^m}$, there exists a $Q'_\epsilon \in \mathcal{Q}_{S'}$ such that $\|Q_\epsilon - Q'_\epsilon\|_1 \leq 2\beta$. Thus, we have

$$
\Psi(S, S') - \Psi(S', S') \leq \langle Q_\epsilon, \ell \rangle - \langle Q_\epsilon, \hat{\ell}_{S'} \rangle + \epsilon - \langle Q'_\epsilon, \ell \rangle - \langle Q'_\epsilon, \hat{\ell}_{S'} \rangle + \epsilon
$$

$$
= \langle Q_\epsilon - Q'_\epsilon, \ell - \hat{\ell}_{S'} \rangle + \epsilon
$$

$$
\leq \|Q_\epsilon - Q'_\epsilon\|_1 \|\ell - \hat{\ell}_{S'}\|_\infty + \epsilon
$$

$$
\leq 2\beta \sup_h |\ell(h) - \hat{\ell}_{S'}(h)| + \epsilon.
$$

Since this bound holds for any $\epsilon > 0$, we conclude that $\Psi(S, S') - \Psi(S', S') \leq 2\beta \sup_h |\ell(h) - \hat{\ell}_{S'}(h)|$, which implies that

$$
\Psi(S, S) - \Psi(S', S') \leq 2\beta \sup_h |\ell(h) - \hat{\ell}_{S'}(h)| + \frac{1}{m} \leq 2\beta + \frac{1}{m}.
$$

Now, via standard Rademacher complexity bounds Mohri et al. [2018], with probability at least $1 - \delta$ over the choice of $S'$, we have

$$
\sup_h |\ell(h) - \hat{\ell}_{S'}(h)| \leq 2 \mathfrak{R}_m(\mathcal{H}) + \sqrt{\frac{\log(2/\delta)}{2m}}.
$$

Thus, with probability at least $1 - \delta'$ over the choice of $S'$, we have

$$
\Psi(S, S) - \Psi(S', S') \leq 2\beta \left( 2 \mathfrak{R}_m(\mathcal{H}) + \sqrt{\frac{\log(2/\delta')}{2m}} \right) + \frac{1}{m}.
$$

Define $B \coloneqq 2\beta\left(2\Re_m(\mathcal{H}) + \sqrt{\frac{\log(2/\delta')}{2m}}\right) + \frac{1}{m}$ for notational convenience. Now we can apply a variant of McDiarmid's inequality that allow almost-everywhere stability [Kutin and Niyogi, 2002] (using the explicit form in Theorem 5.2 in [Rakhlin et al., 2005] with $M = 2\beta + \frac{1}{m}$, $\beta_n = B$, and $\delta_n = \delta'$) to conclude that for any $t > 0$,

$$\mathbb{P}[|\Psi(S,S) - \mathbb{E}\,\Psi(S,S)| \geq t] \leq 2\exp\left(\frac{-t^2}{8nB^2}\right) + \frac{2(2\beta + \frac{1}{m})m\delta'}{B} \leq 2\exp\left(\frac{-t^2}{8nB^2}\right) + 2m^{1.5}\delta'.$$

Now, set $\delta' = \frac{\delta}{2m^{1.5}}$ and $t = B\sqrt{8m\log(\frac{4}{\delta})}$ so that $\mathbb{P}[|\Psi(S,S) - \mathbb{E}\,\Psi(S,S)| \geq t] \leq \delta$. Finally, exactly as in [Foster et al., 2019], we have $\mathbb{E}_{S\sim\mathcal{D}^m}[\Psi(S,S)] \leq 2\Re_m^\diamond(\mathcal{Q}_m)$. $\qquad\qquad\square$

## B.2 Explicit bound of Theorem 4

**Theorem 4.** *Suppose the family of sample-dependent priors $(P_S)_{S\in\mathcal{Z}^m}$ has $\mathsf{D}_\infty$ sensitivity $\epsilon$. Also assume that for some $\eta > 0$, we have $P_S(h) \geq \eta$ for all $h \in \mathcal{H}$, and all $S \in \mathcal{Z}^m$. Then, for any $\delta > 0$, with probability at least $1 - \delta$ over the draw of the sample $S \sim \mathcal{D}^m$, the following inequality holds for all $Q \in \Delta(\mathcal{H})$: if $D = \max\{\mathsf{D}(Q\|P_S), 2\}$,*

$$\mathop{\mathbb{E}}_{\substack{h\sim Q \\ z\sim\mathcal{D}}}[L(h,z)] \leq \mathop{\mathbb{E}}_{h\sim Q}\left[\frac{1}{m}\sum_{i=1}^m L(h,z_i)\right] + 2\sqrt{\frac{4D}{m} + 2\epsilon^2 + 2\epsilon\sqrt{\frac{\log(2m^2/\eta)}{m}}} + \sqrt{\frac{8}{m}} + \frac{2}{m}$$

$$+ \left(4\epsilon\left(2\Re_m(\mathcal{H}) + \sqrt{\frac{\log(4m^{1.5}D/\delta)}{2m}}\right) + \frac{1}{m}\right)\sqrt{8m\log\left(\frac{4D}{\delta}\right)}.$$

## B.3 Lemma 9 & Proof

**Lemma 9** (Extension of Lemma 3.17 in [Dwork and Roth, 2014]). *Let $\mathcal{P}$ be a distribution on $(S,T,h)$ s.t. $\mathsf{D}_\infty^\gamma(\mathcal{P} \parallel \mathcal{D}^{2m} \otimes \mathcal{P}) \leq \kappa$, where $\mathcal{D}^{2m}$ is the marginal distribution of $(S,T)$ induced by $\mathcal{P}$ and $\mathcal{P}$ is the marginal distribution of $h$ induced by $\mathcal{P}$. Then $\exists$ a distribution $\mathcal{P}'$ on $(S,T,h)$ s.t. $\|\mathcal{P} - \mathcal{P}'\|_{\mathrm{TV}} \leq \gamma$ and $\mathsf{D}_\infty(\mathcal{P}' \parallel \mathcal{D}^{2m} \otimes \mathcal{P}) \leq \kappa$ (following Lemma 3.17) and, further, $\mathcal{P}$ and $\mathcal{P}'$ induce the same marginal distributions on $(S,T)$ - i.e., the marginal distribution of $(S,T)$ induced by $\mathcal{P}'$ is also $\mathcal{D}^{2m}$.*

*Proof.* We construct $\mathcal{P}'$ s.t. $\mathcal{P}'_{S,T} = \mathcal{D}^{2m}$ (i.e., the marginal distribution of $(S,T)$ matches that of $\mathcal{P}$ by design) and then, for any fixed $(S,T)$, we define the conditional distribution $\mathcal{P}'_{h|(S,T)}$ in terms of $\mathcal{P}_{h|(S,T)}$ as follows (as is done in Lemma 3.17):

Let $\mathsf{S}_{S,T} \coloneqq \{h : \mathcal{P}_{h|(S,T)}(h) > e^\kappa \cdot \mathcal{P}(h)\}$ and $\mathsf{T}_{S,T} \coloneqq \{h : \mathcal{P}_{h|(S,T)}(h) < \mathcal{P}(h)\}$. (For the moment, $\kappa$ can be thought of as any positive constant; its connection to our assumption will only come into play at the end, with $\gamma$.)

We want to remove the following total probability from $\mathsf{S}_{S,T}$:

$$\sum_{h\in\mathsf{S}_{S,T}}\left[\mathcal{P}_{h|(S,T)}(h) - e^\kappa \cdot \mathcal{P}(h)\right] = \mathcal{P}_{h|(S,T)}(\mathsf{S}_{S,T}) - e^\kappa \cdot \mathcal{P}(\mathsf{S}_{S,T})$$

And we have the following additional capacity in $\mathsf{T}_{S,T}$:

$$\sum_{h\in\mathsf{T}_{S,T}}\left[\mathcal{P}(h) - \mathcal{P}_{h|(S,T)}(h)\right] = \sum_{h\notin\mathsf{T}_{S,T}}\left[\mathcal{P}_{h|(S,T)}(h) - \mathcal{P}(h)\right]$$

$$\geq \sum_{h\in\mathsf{S}_{S,T}}\left[\mathcal{P}_{h|(S,T)}(h) - \mathcal{P}(h)\right]$$

$$\geq \sum_{h\in\mathsf{S}_{S,T}}\left[\mathcal{P}_{h|(S,T)}(h) - e^\kappa \cdot \mathcal{P}(h)\right],$$

which exceeds the mass we want to remove from $\mathsf{S}_{S,T}$.

Therefore, just as in Lemma 3.17, we can lower the probabilities for $h \in \mathsf{S}_{S,T}$ and raise the probabilities for $h \in \mathsf{T}_{S,T}$ to construct $\mathcal{P}'_{h|(S,T)}$. We obtain:

1. $\forall h \in \mathsf{S}_{S,T}, \mathcal{P}'_{h|(S,T)}(h) = e^{\kappa} \cdot \mathcal{P}(h) < \mathcal{P}_{h|(S,T)}(h).$

2. $\forall h \in \mathsf{T}_{S,T}, \mathcal{P}_{h|(S,T)}(h) \le \mathcal{P}'_{h|(S,T)}(h) \le \mathcal{P}(h).$

3. $\forall h \notin \mathsf{S}_{S,T} \cup \mathsf{T}_{S,T}, \mathcal{P}'_{h|(S,T)}(h) = \mathcal{P}_{h|(S,T)}(h) \le e^{\kappa} \cdot \mathcal{P}(h).$

We thus have $\mathsf{D}_{\infty}(\mathcal{P}'_{h|(S,T)} \parallel \mathcal{P}) \le \kappa$ and consequently $\mathsf{D}_{\infty}(\mathcal{P}' \parallel \mathcal{D}^{2m} \otimes \mathcal{P}) \le \kappa$, due to the equivalent marginal distributions on $(S,T)$.

Formally, our original assumption $\mathsf{D}_{\infty}^{\gamma}(\mathcal{P} \parallel \mathcal{D}^{2m} \otimes \mathcal{P}) \le \kappa$ means that for all events $E$:

$$\mathcal{P}(E) - e^{\kappa} \cdot (\mathcal{D}^{2m} \otimes \mathcal{P})(E) \le \gamma.$$

Let $E := \{(S,T,h) \in \mathcal{D}^{2m} \times \mathcal{H} : \mathcal{P}_{h|(S,T)}(h) > e^{\kappa} \cdot \mathcal{P}(h)\}$. We then have:

$$\begin{aligned}
\|\mathcal{P}' - \mathcal{P}\|_{\mathrm{TV}} &= \mathop{\mathbb{E}}_{(S,T)\sim\mathcal{D}^{2m}} \left[ \|\mathcal{P}'_{h|(S,T)} - \mathcal{P}_{h|(S,T)}\|_{\mathrm{TV}} \right] \\
&= \mathop{\mathbb{E}}_{(S,T)\sim\mathcal{D}^{2m}} \left[ \mathcal{P}_{h|(S,T)}(\mathsf{S}_{S,T}) - \mathcal{P}'_{h|(S,T)}(\mathsf{S}_{S,T}) \right] \\
&= \mathop{\mathbb{E}}_{(S,T)\sim\mathcal{D}^{2m}} \left[ \mathcal{P}_{h|(S,T)}(\mathsf{S}_{S,T}) - e^{\kappa} \cdot \mathcal{P}(\mathsf{S}_{S,T}) \right] \\
&= \mathop{\mathbb{E}}_{(S,T)\sim\mathcal{D}^{2m}} \left[ \mathcal{P}(E|S,T) - e^{\kappa} \cdot (\mathcal{D}^{2m} \otimes \mathcal{P})(E|S,T) \right] \\
&= \mathcal{P}(E) - e^{\kappa} \cdot (\mathcal{D}^{2m} \otimes \mathcal{P})(E) \\
&\le \gamma.
\end{aligned}$$

We have thus shown that $\|\mathcal{P}' - \mathcal{P}\|_{\mathrm{TV}} \le \gamma$ and $\mathsf{D}_{\infty}(\mathcal{P}' \parallel \mathcal{D}^{2m} \otimes \mathcal{P}) \le \kappa$ for a $\mathcal{P}'$ whose marginal distribution on $(S,T)$ matches that of $\mathcal{P}$. $\qquad\square$

### B.4  Proof of Theorem 5

We prove Theorem 5, with the exact bound explicitly spelled out:

**Theorem 5.** *Suppose the family of sample-dependent priors $(P_S)_{S\in\mathcal{Z}^m}$ has $\mathsf{D}_{\infty}$ sensitivity $\epsilon$. Then, for any $\delta > 0$, with probability at least $1 - \delta$ over the draw of the sample $S \sim \mathcal{D}^m$, the following inequality holds for all $Q \in \Delta(\mathcal{H})$: if $D = \max\{\mathsf{D}(Q\|P_S), 2\}$,*

$$\begin{aligned}
\mathop{\mathbb{E}}_{\substack{h\sim Q\\z\sim\mathcal{D}}}[L(h,z)] \le{}& \mathop{\mathbb{E}}_{h\sim Q}\left[ \frac{1}{m} \sum_{i=1}^{m} L(h,z_i) \right] \\
& + \max\left\{ 4\sqrt{\frac{4D + 4\log(2)}{m} + 2\epsilon^2 + 2\epsilon\sqrt{\frac{\log(2)}{m}}}, 8\epsilon^{2/3}\mathfrak{R}_m(\mathcal{H})^{1/3}, 8\epsilon^{4/5} \right\} \\
& + \frac{2}{\sqrt{m}} + \left( 4\epsilon\left( 2\mathfrak{R}_m(\mathcal{H}) + \sqrt{\frac{\log(4m^{1.5}D/\delta)}{2m}} \right) + \frac{1}{m} \right) \sqrt{8m\log\left(\tfrac{4D}{\delta}\right)}.
\end{aligned}$$

*Proof.* Define a sample-dependent family of distributions $\mathcal{Q}_m = (\mathcal{Q}_S)_{S\in\mathcal{Z}^m}$ where $\mathcal{Q}_S = \{Q: \mathsf{D}_{\infty}(Q\|P_S) \le \mu\}$ for some parameter $\mu$. We now apply the bound in Theorem 3, using the bound on the Rademacher complexity from Lemma 10, and the bound $\beta \le 2\epsilon$ from Lemma 6. Finally, a uniform bound over all values of $\mu$ follows by an application of Lemma 3. $\qquad\square$

**Lemma 10.** *If $\mathsf{D}_{\infty}(P_S \parallel P_{S'}) \le \epsilon$ for all $S, S' \in \mathcal{Z}^m$ differing by exactly one point, then*

$$\mathfrak{R}_m^{\diamond}(\mathcal{Q}_{m,\mu}) \le \max\left\{ 2\sqrt{\frac{2\mu + 4\log(2)}{m} + 2\epsilon^2 + 2\epsilon\sqrt{\frac{\log(2)}{m}}}, 4\epsilon^{2/3}\mathfrak{R}_m(\mathcal{H})^{1/3}, 4\epsilon^{4/5} \right\} + \frac{1}{\sqrt{m}}.$$

*Proof.* Assume $\mathsf{D}_\infty(P_S \parallel P_{S'}) \leq \epsilon$ for all $S, S' \in \mathcal{Z}^m$ differing by exactly one point.

Now, we fix the value of $\boldsymbol{\sigma} \in \{-1, 1\}^m$ and introduce the following two distributions on $\mathcal{H}$:

(1) Let $\mathcal{P}_{\boldsymbol{\sigma}}$ be a joint distribution on $(S, T, h)$ induced by sampling $S, T \sim \mathcal{D}^m$, and then, conditioned on the values of $S$ and $T$, sampling $h \sim P_{S_T^{\boldsymbol{\sigma}}}$, using the notation $P_{S_T^{\boldsymbol{\sigma}}}$ introduced for Equation 8.

(2) Let $\mathcal{P}$ be the marginal distribution of $h$ induced by $\mathcal{P}_{\boldsymbol{\sigma}}$. We have dropped $\boldsymbol{\sigma}$ from the notation because - since all elements of $S$ and $T$ are sampled i.i.d. - we have:

$$\mathbb{E}_{S,T\sim\mathcal{D}^m}\big[P_{S_T^{\boldsymbol{\sigma}}}(h)\big] = \mathbb{E}_{S\sim\mathcal{D}^m}\big[P_S(h)\big],$$

i.e., the marginal distribution of $h$ is independent of $\boldsymbol{\sigma}$.

We first invoke several differential privacy results to show that, for the distributions $\mathcal{P}_{\boldsymbol{\sigma}}$ and $\mathcal{P}$ as defined above, and $\kappa := \epsilon^2 m + \epsilon\sqrt{m\log(2/\gamma)}$, we have:

$$\mathsf{D}_\infty^\gamma(\mathcal{P}_{\boldsymbol{\sigma}} \parallel \mathcal{D}^{2m} \otimes \mathcal{P}) \leq \kappa. \tag{14}$$

Specifically, consider $U = (S, T)$ and $U' = (S', T')$ for $S, T, S', T' \in \mathcal{Z}^m$ such that $U$ and $U'$ differ by only *one* of their $2m$ elements. Then $S_T^{\boldsymbol{\sigma}}$ and $S'^{\boldsymbol{\sigma}}_{T'}$ can only differ by at most one element, so by our main assumption: $\mathsf{D}_\infty(P_{S_T^{\boldsymbol{\sigma}}} \parallel P_{S'^{\boldsymbol{\sigma}}_{T'}}) \leq \epsilon$. Crucially, another way of saying this is: the algorithm $\mathcal{A}$ taking $U = (S, T)$ as input and outputting $h \sim P_{S_T^{\boldsymbol{\sigma}}}$ is an $\epsilon$-differentially private algorithm, so we can apply Theorem 20 in [Dwork et al., 2015], with an input of size $2m$, and obtain (14).

We now use Lemma 3.17 (Part 1) in [Dwork and Roth, 2014] to convert (14) into a result concerning $\mathsf{D}_\infty$ vs. $\mathsf{D}_\infty^\gamma$, so we can more easily use it below. Specifically, by Lemma 3.17 (Part 1), there exists a distribution $\mathcal{P}'_{\boldsymbol{\sigma}}$ on $(S, T, h)$ such that $\|\mathcal{P}_{\boldsymbol{\sigma}} - \mathcal{P}'_{\boldsymbol{\sigma}}\|_{\text{TV}} \leq \gamma$ and $\mathsf{D}_\infty(\mathcal{P}'_{\boldsymbol{\sigma}} \parallel \mathcal{D}^{2m} \otimes \mathcal{P}) \leq \kappa$.

Finally, we upper bound $\mathfrak{R}_m^\diamond(\mathcal{Q}_{m,\mu})$ as follows. For convenience, we use a variable $t > 0$ and the function $\Psi_P(Q)$, which is defined as $\mathsf{D}(Q \parallel P)$ if $\mathsf{D}(Q \parallel P) \leq \mu$ and $+\infty$ otherwise; thus, its conjugate function is $\Psi_P^*(u) = \log\big(\mathbb{E}_{h\in P}[e^{u(h)}]\big)$, for all $u \in \mathbb{R}^{\mathcal{H}}$ [Mohri et al., 2018, Lemma B.37]. We use the shorthand $u_{\boldsymbol{\sigma}}(h) = \sum_{i=1}^m \sigma_i L(h, z_i)$, where $z_i$ is element $i$ of sample $T$, so that $\sum_{i=1}^m \sigma_i \langle Q, L_{z_i} \rangle = \langle Q, u_{\boldsymbol{\sigma}} \rangle$.

$$\begin{aligned}
\mathfrak{R}_m^\diamond(\mathcal{Q}_{m,\mu}) &= \frac{1}{mt} \mathbb{E}_{\boldsymbol{\sigma}} \mathbb{E}_{(S,T)} \left[ \sup_{\mathsf{D}(Q\|P_{S_T^{\boldsymbol{\sigma}}})\leq\mu} \langle Q, tu_{\boldsymbol{\sigma}} \rangle \right] \\
&\leq \frac{1}{mt} \mathbb{E}_{\boldsymbol{\sigma}} \mathbb{E}_{(S,T)} \left[ \sup_{\Psi_{P_{S_T^{\boldsymbol{\sigma}}}}(Q)\leq\mu} \Psi_{P_{S_T^{\boldsymbol{\sigma}}}}(Q) + \Psi_{P_{S_T^{\boldsymbol{\sigma}}}}^*(tu_{\boldsymbol{\sigma}}) \right] \quad \text{(Fenchel inequality)} \\
&\leq \frac{\mu}{mt} + \frac{1}{mt} \mathbb{E}_{\boldsymbol{\sigma}} \mathbb{E}_{(S,T)} \left[ \Psi_{P_{S_T^{\boldsymbol{\sigma}}}}^*(tu_{\boldsymbol{\sigma}}) \right] \\
&= \frac{\mu}{mt} + \frac{1}{mt} \mathbb{E}_{\boldsymbol{\sigma}} \mathbb{E}_{(S,T)} \left[ \log\Big( \mathbb{E}_{h\sim P_{S_T^{\boldsymbol{\sigma}}}} \big[ e^{tu_{\boldsymbol{\sigma}}(h)} \big] \Big) \right] \quad \text{(definition of $\Psi^*$)} \\
&\leq \frac{\mu}{mt} + \frac{1}{mt} \mathbb{E}_{\boldsymbol{\sigma}} \log\Big( \mathbb{E}_{(S,T,h)\sim\mathcal{P}_{\boldsymbol{\sigma}}} \big[ e^{tu_{\boldsymbol{\sigma}}(h)} \big] \Big) \quad \text{(Jensen's inequality)} \tag{15}
\end{aligned}$$

In the following, to make the dependence of $u_{\boldsymbol{\sigma}}$ on the set $T$ explicit, we now denote it as $u_{\boldsymbol{\sigma},T}$. For any sample $T$, define $\Psi(T)$ by $\Psi(T) = \frac{1}{m} \sup_{h\in\mathcal{H}} \big( u_{\boldsymbol{\sigma},T}(h) - \mathbb{E}_{T'\sim\mathcal{D}^m}[u_{\boldsymbol{\sigma},T'}(h)] \big)$. Changing one point in $T$ affects $\Psi(T)$ by at most $1/m$, since the loss is bounded by one. Thus, by McDiarmid's inequality, for any fixed $\boldsymbol{\sigma}$ and for any $\delta > 0$, we have

$$\mathbb{P}_{T\sim\mathcal{D}^m}\left[ \Psi(T) \leq \mathbb{E}_{T\sim\mathcal{D}^m}[\Psi(T)] + \sqrt{\frac{2\log(\frac{1}{\delta})}{m}} \right] \geq 1 - \delta.$$

Now, $\mathbb{E}_{T\sim\mathcal{D}^m}[\Psi(T)]$ can be bounded in terms of the Rademacher complexity as in the standard analyses:

$$
\begin{aligned}
\mathop{\mathbb{E}}_{T\sim\mathcal{D}^m}[\Psi(T)] &= \frac{1}{m}\mathop{\mathbb{E}}_{T\sim\mathcal{D}^m}\left[\sup_{h\in\mathcal{H}}\mathop{\mathbb{E}}_{T'\sim\mathcal{D}^m}[u_{\boldsymbol{\sigma},T}(h)-u_{\boldsymbol{\sigma},T'}(h)]\right]\\
&\leq \frac{1}{m}\mathop{\mathbb{E}}_{T,T'\sim\mathcal{D}^m}\left[\sup_{h\in\mathcal{H}}u_{\boldsymbol{\sigma},T}(h)-u_{\boldsymbol{\sigma},T'}(h)\right] \qquad\text{(sub-additivity of sup)}\\
&\leq \frac{1}{m}\mathop{\mathbb{E}}_{T,T'\sim\mathcal{D}^m}\left[\sup_{h\in\mathcal{H}}\sum_{i=1}^m\left(\sigma_i L(h,z_i^T)-\sigma_i L(h,z_i^{T'})\right)\right]\\
&\leq \frac{1}{m}\mathop{\mathbb{E}}_{T,T'\sim\mathcal{D}^m,\boldsymbol{\beta}}\left[\sup_{h\in\mathcal{H}}\sum_{i=1}^m\beta_i\left(\sigma_i L(h,z_i^T)-\sigma_i L(h,z_i^{T'})\right)\right] \quad\text{(Rademacher variables }\beta_i)\\
&\leq \frac{2}{m}\mathop{\mathbb{E}}_{T\sim\mathcal{D}^m,\boldsymbol{\beta}}\left[\sup_{h\in\mathcal{H}}\sum_{i=1}^m\beta_i\left(\sigma_i L(h,z_i^T)\right)\right]\\
&= \frac{2}{m}\mathop{\mathbb{E}}_{T\sim\mathcal{D}^m,\boldsymbol{\beta}}\left[\sup_{h\in\mathcal{H}}\sum_{i=1}^m\beta_i L(h,z_i^T)\right]\\
&= 2\mathfrak{R}_m(\mathcal{H}).
\end{aligned}
$$

Thus, for any fixed $\boldsymbol{\sigma}$ and for any $\delta>0$, we have

$$
\mathop{\mathbb{P}}_{T\sim\mathcal{D}^m}\left[\sup_h\left(u_{\boldsymbol{\sigma},T}(h)-\mathop{\mathbb{E}}_{T'\sim\mathcal{D}^m}[u_{\boldsymbol{\sigma},T'}(h)]\right)\leq 2m\mathfrak{R}_m(\mathcal{H})+\sqrt{2m\log(1/\delta)}\right]\geq 1-\delta. \tag{16}
$$

Note that for any $h$, we have $\mathbb{E}_{T'\sim\mathcal{D}^m}[u_{\boldsymbol{\sigma},T'}(h)] = \sum_{i=1}^m\sigma_i\mathbb{E}_{z\sim D}[L(h,z)]$, and hence $|\mathbb{E}_{T'\sim\mathcal{D}^m}[u_{\boldsymbol{\sigma},T'}(h)]|\leq|\sum_{i=1}^m\sigma_i|$. Hence, we conclude that

$$
\mathop{\mathbb{P}}_{T\sim\mathcal{D}^m}\left[\sup_h u_{\boldsymbol{\sigma},T}(h)\leq\left|\sum_{i=1}^m\sigma_i\right|+2m\mathfrak{R}_m(\mathcal{H})+\sqrt{2m\log(1/\delta)}\right]\geq 1-\delta. \tag{17}
$$

For notational convenience, define

$$
B_{\boldsymbol{\sigma}}:=\left|\sum_{i=1}^m\sigma_i\right|+2m\mathfrak{R}_m(\mathcal{H})+\sqrt{2m\log(1/\delta)}.
$$

Now, let $\delta:=e^{-tm}$, and let $G\subseteq\mathcal{Z}^m$ be the set of $m$-element samples $T$ such that

$$
G:=\left\{T\in\mathcal{Z}^m:\sup_h u_{\boldsymbol{\sigma},T}(h)\leq B_{\boldsymbol{\sigma}}\right\}.
$$

By (16), we have $\mathbb{P}_{T\sim\mathcal{D}^m}[G]\geq 1-\delta$. Hence, we have

$$
\begin{aligned}
\mathop{\mathbb{E}}_{(S,T,h)\sim\mathcal{P}_{\boldsymbol{\sigma}}}\left[e^{tu_{\boldsymbol{\sigma},T}(h)}\right] &\leq \mathop{\mathbb{E}}_{(S,T,h)\sim\mathcal{P}'_{\boldsymbol{\sigma}}}\left[e^{tu_{\boldsymbol{\sigma},T}(h)}\right]+\left(\sup_{T\in G}\sup_h e^{tu_{\boldsymbol{\sigma},T}(h)}\right)\cdot\left(\left|\mathop{\mathbb{P}}_{\mathcal{P}_{\boldsymbol{\sigma}}}[T\in G]-\mathop{\mathbb{P}}_{\mathcal{P}'_{\boldsymbol{\sigma}}}[T\in G]\right|\right)\\
&\quad + e^{tm}\cdot\mathop{\mathbb{P}}_{\mathcal{P}_{\boldsymbol{\sigma}}}[T\notin G]\\
&\leq \mathop{\mathbb{E}}_{(S,T,h)\sim\mathcal{P}'_{\boldsymbol{\sigma}}}\left[e^{tu_{\boldsymbol{\sigma},T}(h)}\right]+\gamma e^{tB_{\boldsymbol{\sigma}}}+e^{tm}\delta\\
&= \mathop{\mathbb{E}}_{(S,T,h)\sim\mathcal{P}'_{\boldsymbol{\sigma}}}\left[e^{tu_{\boldsymbol{\sigma},T}(h)}\right]+\gamma e^{tB_{\boldsymbol{\sigma}}}+1\\
&\leq \left(\mathop{\mathbb{E}}_{(S,T,h)\sim\mathcal{P}'_{\boldsymbol{\sigma}}}\left[e^{tu_{\boldsymbol{\sigma},T}(h)}\right]+1\right)\cdot\left(\gamma e^{tB_{\boldsymbol{\sigma}}}+1\right).
\end{aligned}
$$

Using this bound in (15), we get

$$
\mathfrak{R}_m^\diamond(\mathcal{Q}_{m,\mu})\leq\frac{\mu}{mt}+\frac{1}{mt}\mathop{\mathbb{E}}_{\boldsymbol{\sigma}}\left[\log\left(\mathop{\mathbb{E}}_{(S,T,h)\sim\mathcal{P}'_{\boldsymbol{\sigma}}}\left[e^{tu_{\boldsymbol{\sigma}}(h)}\right]+1\right)+\log\left(\gamma e^{tB_{\boldsymbol{\sigma}}}+1\right)\right]. \tag{18}
$$

We bound the two terms involving the logarithm in (18) separately. First, we have

$$\mathbb{E}_{\boldsymbol{\sigma}} \log \left( \mathbb{E}_{(S,T,h)\sim \mathcal{P}'_{\boldsymbol{\sigma}}} \left[ e^{tu_{\boldsymbol{\sigma}}(h)} \right] + 1 \right)$$

$$\leq \mathbb{E}_{\boldsymbol{\sigma}} \log \left( \mathbb{E}_{(S,T,h)\sim \mathcal{D}^{2m}\otimes \mathcal{P}} \left[ e^{\kappa} e^{tu_{\boldsymbol{\sigma}}(h)} \right] + 1 \right) \quad \text{(since } \mathsf{D}_{\infty}(\mathcal{P}'_{\boldsymbol{\sigma}} \| \mathcal{D}^{2m}\otimes \mathcal{P}) \leq \kappa\text{)}$$

$$\leq \log \left( \mathbb{E}_{(S,T,h)\sim \mathcal{D}^{2m}\otimes \mathcal{P}} \mathbb{E}_{\boldsymbol{\sigma}} \left[ e^{\kappa} e^{tu_{\boldsymbol{\sigma}}(h)} \right] + 1 \right) \quad \text{(Jensen's inequality)}$$

$$\leq \log \left( \mathbb{E}_{(S,T,h)\sim \mathcal{D}^{2m}\otimes \mathcal{P}} e^{\kappa+mt^2/2} + 1 \right) \quad \text{(Hoeffding's lemma)}$$

$$\leq \log \left( 2 e^{\kappa+mt^2/2} \right) \quad (e^{k+mt^2/2} \geq 1)$$

$$\leq \kappa + \frac{mt^2}{2} + \log(2). \tag{19}$$

As for the second term, setting $\gamma = e^{-(2mt\mathfrak{R}_m(\mathcal{H})+\sqrt{2}mt^{3/2})}$, we have

$$\mathbb{E}_{\boldsymbol{\sigma}} \log \left( \gamma e^{tB_{\boldsymbol{\sigma}}} + 1 \right) = \mathbb{E}_{\boldsymbol{\sigma}} \log \left( \gamma e^{t(|\sum_{i=1}^m \sigma_i| + 2m\mathfrak{R}_m(\mathcal{H}) + \sqrt{2m\log(1/\delta)})} + 1 \right) \quad \text{(definition of } B_{\boldsymbol{\sigma}}\text{)}$$

$$= \mathbb{E}_{\boldsymbol{\sigma}} \log \left( e^{t|\sum_{i=1}^m \sigma_i|} + 1 \right) \quad \left( \text{using } \gamma = e^{-(2mt\mathfrak{R}_m(\mathcal{H})+\sqrt{2}mt^{3/2})} \right)$$

$$\leq \mathbb{E}_{\boldsymbol{\sigma}} \log \left( 2 e^{t|\sum_{i=1}^m \sigma_i|} \right)$$

$$= \mathbb{E}_{\boldsymbol{\sigma}} \left[ t \Big| \sum_{i=1}^m \sigma_i \Big| \right] + \log(2)$$

$$= t \mathbb{E}_{\boldsymbol{\sigma}} \left[ \sqrt{(\sum_{i=1}^m \sigma_i)^2} \right] + \log(2)$$

$$\leq t \sqrt{ \mathbb{E}_{\boldsymbol{\sigma}} \left[ (\sum_{i=1}^m \sigma_i)^2 \right] } + \log(2) \quad \text{(Jensen's inequality)}$$

$$= \sqrt{m} t + \log(2) \tag{20}$$

Using bounds (19), (20), and the bound on $k$ in (18) we get

$$\mathfrak{R}_m^{\diamond}(\mathcal{Q}_{m,\mu}) \leq \frac{1}{mt} \left( \mu + \kappa + \frac{mt^2}{2} + \sqrt{m} t + 2\log(2) \right)$$

$$\leq \frac{1}{mt} \left( \mu + \epsilon^2 m + \epsilon \sqrt{m(2mt\mathfrak{R}_m(\mathcal{H}) + \sqrt{2}mt^{3/2})} + m\log(2) + \frac{mt^2}{2} + \sqrt{m} t + 2\log(2) \right)$$

$$\leq \max \left\{ 2\sqrt{ \frac{2\mu + 4\log(2)}{m} + 2\epsilon^2 + 2\epsilon\sqrt{\frac{\log(2)}{m}} }, 4\epsilon^{2/3}\mathfrak{R}_m(\mathcal{H})^{1/3}, 4\epsilon^{4/5} \right\} + \frac{1}{\sqrt{m}},$$

setting $t = \max \left\{ \sqrt{ \frac{2\mu + 4\log(2)}{m} + 2\epsilon^2 + 2\epsilon\sqrt{\frac{\log(2)}{m}} }, 2\epsilon^{2/3}\mathfrak{R}_m(\mathcal{H})^{1/3}, 2\epsilon^{4/5} \right\}$. $\qquad \square$

### B.5  Proof of Theorem 6

The requirement in Theorem 5 that the family of sample-dependent priors $(P_S)_{S\in\mathcal{Z}^m}$ has $\mathsf{D}_{\infty}$ sensitivity $\epsilon$ is equivalent to saying that the priors define an $\epsilon$-differentially private mechanism. Here, we give an extension to Theorem 5 which makes the weaker assumption that the priors define an $(\epsilon, \delta)$-differentially private mechanism, for some $\delta > 0$. The extension relies on the following theorem of Rogers et al. [2016]. The statement given below is an adaptation of Theorem 3.1 in [Rogers et al., 2016] that is implicit in their proof. We need this more nuanced statement for our analysis.

**Theorem 7** (Theorem 3.1 in [Rogers et al., 2016]). *Let $\mathcal{A} : \mathcal{X}^m \to \mathcal{Y}$ be an $(\epsilon, \delta)$-differentially private algorithm for $\epsilon \in (0, \frac{1}{2}]$ and $\delta \in (0, \epsilon)$. Let $\mathcal{D}$ be any distribution on $\mathcal{X}$ and let $S \in \mathcal{X}^m$ be a dataset with elements sampled i.i.d. from $\mathcal{D}$. Let $\mathcal{P}$ be the joint distribution of $(S, \mathcal{A}(S))$, and $P$ be the marginal distribution of $\mathcal{A}(S)$. Then there is a constant $c > 0$ such that for any $\gamma \in (0, 1]$ we have*

$$\mathsf{D}_{\infty}^{\delta + c\sqrt{\frac{\delta}{\epsilon}}m}(\mathcal{P} \| \mathcal{D}^m \otimes P) \leq 72\epsilon^2 m + 6\epsilon\sqrt{2m\log(1/\gamma)} + c\sqrt{\frac{\delta}{\epsilon}}m.$$

With this theorem, we can now prove the following theorem which is analogous to Theorem 5 but assumes only the priors define an $(\epsilon, \delta)$-differentially private mechanism.

**Theorem 6.** *Assume that $\epsilon \geq 0$ and $\delta \in [0, \frac{e^{-16m}}{4c^2m^2}\epsilon]$, where $c$ is the constant from Theorem 7. Suppose the family of sample-dependent priors $(P_S)_{S \in \mathcal{Z}^m}$ satisfy the property that $\mathsf{D}_\infty^\delta(P_S\|P_{S'}) \leq \epsilon$ for all $S, S' \in \mathcal{Z}^m$ differing in exactly one point. Then, for any $\nu > 0$, with probability at least $1 - \nu$ over the draw of the sample $S \sim \mathcal{D}^m$, the following inequality holds for all $Q \in \Delta(\mathcal{H})$: if $D = \max\{\mathsf{D}(Q\|P_S), 2\}$,*

$$
\mathop{\mathbb{E}}_{\substack{h \sim Q \\ z \sim \mathcal{D}}}[L(h,z)] \leq \mathop{\mathbb{E}}_{h \sim Q}\left[\frac{1}{m}\sum_{i=1}^m L(h, z_i)\right]
$$

$$
+ \max\left\{4\sqrt{\frac{4D + 6\log(2)}{m} + 300\epsilon^2}, 30\epsilon^{2/3}\mathfrak{R}_m(\mathcal{H})^{1/3}, 30\epsilon^{4/5}\right\}
$$

$$
+ \frac{2}{\sqrt{m}} + \frac{c\sqrt{\delta}}{4\epsilon^{3/2}} + \left(4\epsilon\left(2\mathfrak{R}_m(\mathcal{H}) + \sqrt{\frac{\log(4m^{1.5}D/\nu)}{2m}}\right) + \frac{1}{m}\right)\sqrt{8m\log\left(\frac{4D}{\nu}\right)}.
$$

*Proof.* Define a sample-dependent family of distributions $\mathcal{Q}_m = (\mathcal{Q}_S)_{S \in \mathcal{Z}^m}$ where $\mathcal{Q}_S = \{Q\colon \mathsf{D}_\infty(Q\|P_S) \leq \mu\}$ for some parameter $\mu$. We now apply the bound in Theorem 3, using the bound on the Rademacher complexity from Lemma 11, and the bound $\beta \leq 2\epsilon$ from Lemma 6. Finally, a uniform bound over all values of $\mu$ follows by an application of Lemma 3. $\square$

**Lemma 11.** *Assume that $\epsilon \geq 0$ and $\delta \in [0, \frac{e^{-16m}}{4c^2m^2}\epsilon]$, where $c$ is the constant from Theorem 7. Suppose that $\mathsf{D}_\infty^\delta(P_S\|P_{S'}) \leq \epsilon$ for all $S, S' \in \mathcal{Z}^m$ differing in exactly one point. Then,*

$$
\mathfrak{R}_m^\diamond(\mathcal{Q}_{m,\mu}) \leq \max\left\{2\sqrt{\frac{2\mu + 6\log(2)}{m} + 300\epsilon^2}, 15\epsilon^{2/3}\mathfrak{R}_m(\mathcal{H})^{1/3}, 15\epsilon^{4/5}\right\} + \frac{1}{\sqrt{m}} + \frac{c\sqrt{\delta}}{8\epsilon^{3/2}}.
$$

*Proof.* The proof is exactly along the lines of the proof of Lemma 10. Instead of using Theorem 20 in [Dwork et al., 2015], we use Theorem 7 above. Using this theorem, the proof of Lemma 11 follows with

$$
\kappa = 144\epsilon^2 m + 12\epsilon\sqrt{m\log(1/\gamma)} + 2c\sqrt{\tfrac{\delta}{\epsilon}}m
$$

and $\gamma$ replaced by $\gamma + 2c\sqrt{\frac{\delta}{\epsilon}}m$. The bound (20) changes as follows: setting $\gamma = e^{-(2mt\mathfrak{R}_m(\mathcal{H}) + \sqrt{2}mt^{3/2})}$ exactly as in the proof of Lemma 10, and assuming that we choose $t \leq 2$ ($t > 2$ leads to a trivial bound), we note that $\gamma + 2c\sqrt{\frac{\delta}{\epsilon}}m \leq 2\gamma$ since we assumed that $\delta \leq \frac{e^{-16m}}{4c^2m^2}\epsilon$, and hence

$$
\mathop{\mathbb{E}}_{\sigma}\log\left(\left(\gamma + 2c\sqrt{\tfrac{\delta}{\epsilon}}m\right)e^{tB_\sigma} + 1\right) \leq \mathop{\mathbb{E}}_{\sigma}\log\left(2\gamma e^{tB_\sigma} + 1\right) \leq \sqrt{m}t + \log(4).
$$

Finally, we have

$$
\mathfrak{R}_m^\diamond(\mathcal{Q}_{m,\mu}) \leq \frac{1}{mt}\left(\mu + \kappa + \frac{mt^2}{2} + \sqrt{m}t + 3\log(2)\right)
$$

$$
\leq \frac{1}{mt}\left(\mu + 144\epsilon^2 m + 12\epsilon\sqrt{m(2mt\mathfrak{R}_m(\mathcal{H}) + \sqrt{2}mt^{3/2})} \right) + 2c\sqrt{\frac{\delta}{\epsilon}}m + \frac{mt^2}{2} + \sqrt{m}t
$$

$$
\left. + 3\log(2)\right)
$$

$$
\leq \max\left\{2\sqrt{\frac{2\mu + 6\log(2)}{m} + 300\epsilon^2}, 15\epsilon^{2/3}\mathfrak{R}_m(\mathcal{H})^{1/3}, 15\epsilon^{4/5}\right\} + \frac{1}{\sqrt{m}} + \frac{c\sqrt{\delta}}{8\epsilon^{3/2}},
$$

setting $t = \min\left\{\max\left\{\sqrt{\frac{2\mu + 6\log(2)}{m} + 300\epsilon^2}, 15\epsilon^{2/3}\mathfrak{R}_m(\mathcal{H})^{1/3}, 15\epsilon^{4/5}\right\}, 2\right\}$ and using the bound $\frac{2c}{t}\sqrt{\frac{\delta}{\epsilon}} \leq \frac{2c}{\sqrt{300\epsilon^2}}\sqrt{\frac{\delta}{\epsilon}} \leq \frac{c\sqrt{\delta}}{8\epsilon^{3/2}}$. $\square$

**Remark.** The stipulation that $\delta \leq \frac{e^{-16m}}{4c^2 m^2}\epsilon$ in the statement of Lemma 11 is made simply to yield a clean statement. It should be evident from the proof that other values of $\delta$ also yield analogous bounds on the Rademacher complexity. For example, we can allow $\delta$ to be as large as $\frac{e^{-(4mt\mathfrak{R}_m(\mathcal{H})+2\sqrt{2}mt^{3/2})}}{4c^2 m^2}\epsilon$ for the value of $t$ in the proof above and retain the exact same bound.

## B.6 Proof of Lemma 5

**Lemma 5.** *Suppose $\|P_S - P_{S'}\|_1 \leq \epsilon$ for all $S, S' \in \mathcal{Z}^m$ differing by exactly one point. For some $\mu \geq 0$, define the sample-dependent set of distributions as $\mathcal{Q}_{S,\mu} := \{Q \colon \mathsf{D}(Q\|P_S) \leq \mu\}$, and the corresponding family to be $\mathcal{Q}_{m,\mu} = (\mathcal{Q}_{S,\mu})_{S\in\mathcal{Z}^m}$. Then $\mathcal{Q}_{m,\mu}$ is $\beta$-stable for $\beta = \min\left\{\frac{\epsilon d_\infty}{\sqrt{2\mu}}, \sqrt{\frac{\epsilon d_\infty}{2}}\right\}$, where $d_\infty := \sup_{S,S',Q\in\mathcal{Q}_{S,\mu}}\left\|\frac{Q}{P_{S'}}\right\|_\infty$.*

*Proof.* Consider an arbitrary $Q \in \mathcal{Q}_{S,\mu}$.

Case (1): $\mathsf{D}(Q \| P_{S'}) \leq \mu$.
In this case, $Q \in \mathcal{Q}_{S',\mu}$, so we choose $Q' = Q$, and thus $\|Q' - Q\|_{\mathrm{TV}} = 0$.

Case (2): $\mathsf{D}(Q \| P_{S'}) > \mu$.
We consider $Q' = \lambda Q + (1 - \lambda)P_{S'}$, for $\lambda = \frac{\mathsf{D}(Q\|P_S)}{\mathsf{D}(Q\|P_{S'})} < 1$. We show that $Q' \in \mathcal{Q}_{S',\mu}$ as follows:

$$\mathsf{D}(Q' \| P_{S'}) = \mathsf{D}(\lambda Q + (1 - \lambda)P_{S'} \| P_{S'}) \leq \lambda \mathsf{D}(Q \| P_{S'}) + (1 - \lambda)\mathsf{D}(P_{S'} \| P_{S'}) = \mathsf{D}(Q \| P_S) \leq \mu,$$

where the inequality is by the convexity of relative entropy.

We can upper bound $\|Q' - Q\|_{\mathrm{TV}}$ in two different ways.
One way is to directly upper bound the TV distance as follows:

$$
\begin{aligned}
\|Q' - Q\|_{\mathrm{TV}} &= \|\lambda Q + (1 - \lambda)P_{S'} - Q\|_{\mathrm{TV}} \\
&= (1 - \lambda)\|Q - P_{S'}\|_{\mathrm{TV}} \\
&= \left[1 - \frac{\mathsf{D}(Q \| P_S)}{\mathsf{D}(Q \| P_{S'})}\right]\|Q - P_{S'}\|_{\mathrm{TV}} \\
&= [\mathsf{D}(Q \| P_{S'}) - \mathsf{D}(Q \| P_S)]\frac{\|Q - P_{S'}\|_{\mathrm{TV}}}{\mathsf{D}(Q \| P_{S'})} \\
&\leq \frac{\mathsf{D}(Q \| P_{S'}) - \mathsf{D}(Q \| P_S)}{\sqrt{2\mathsf{D}(Q \| P_{S'})}} \qquad \text{(Pinsker's inequality)}.
\end{aligned}
$$

Alternatively, we can upper bound the TV distance by upper bounding the KL divergence as follows:

$$
\begin{aligned}
\mathsf{D}(Q \| Q') &= \mathsf{D}(Q \| \lambda Q + (1 - \lambda)P_{S'}) \\
&\leq (1 - \lambda)\mathsf{D}(Q \| P_{S'}) \qquad \text{(convexity of relative entropy)} \\
&= \left[1 - \frac{\mathsf{D}(Q \| P_S)}{\mathsf{D}(Q \| P_{S'})}\right]\mathsf{D}(Q \| P_{S'}) \\
&= \mathsf{D}(Q \| P_{S'}) - \mathsf{D}(Q \| P_S) \\
\implies \|Q' - Q\|_{\mathrm{TV}} &\leq \sqrt{\frac{\mathsf{D}(Q \| P_{S'}) - \mathsf{D}(Q \| P_S)}{2}} \qquad \text{(Pinsker's inequality)}.
\end{aligned}
$$

We upper bound the common term $\mathsf{D}(Q \parallel P_{S'}) - \mathsf{D}(Q \parallel P_S)$ as follows:

$$
\begin{aligned}
\mathsf{D}(Q \parallel P_{S'}) - \mathsf{D}(Q \parallel P_S) &= \mathop{\mathbb{E}}_{h \sim Q}\left[\log \frac{Q(h)}{P_{S'}(h)}\right] - \mathop{\mathbb{E}}_{h \sim Q}\left[\log \frac{Q(h)}{P_S(h)}\right] \quad \text{(def. of relative entropy)} \\
&= \mathop{\mathbb{E}}_{h \sim Q}\left[\log \frac{P_S(h)}{P_{S'}(h)}\right] \\
&\leq \mathop{\mathbb{E}}_{h \sim Q}\left[\frac{P_S(h)}{P_{S'}(h)} - 1\right] \quad (\log x \leq x - 1) \\
&= \sum_{h \in \mathcal{H}} Q(h)\left[\frac{P_S(h)}{P_{S'}(h)} - 1\right] \\
&= \sum_{h \in \mathcal{H}} \frac{Q(h)}{P_{S'}(h)}\left[P_S(h) - P_{S'}(h)\right] \\
&\leq \left\|\frac{Q}{P_{S'}}\right\|_\infty \|P_S - P_{S'}\|_1 \quad \text{(Hölder's inequality)} \\
&\leq \epsilon d_\infty\left(\frac{Q}{P_{S'}}\right),
\end{aligned}
$$

where $d_\infty(f) \coloneqq \|f\|_\infty$.

Putting this together, we obtain:

$$
\begin{aligned}
\|Q' - Q\|_{\mathrm{TV}} &\leq \min\left\{\frac{\mathsf{D}(Q \parallel P_{S'}) - \mathsf{D}(Q \parallel P_S)}{\sqrt{2\mathsf{D}(Q \parallel P_{S'})}}, \sqrt{\frac{\mathsf{D}(Q \parallel P_{S'}) - \mathsf{D}(Q \parallel P_S)}{2}}\right\} \\
&\leq \min\left\{\frac{\epsilon}{\sqrt{2\mu}} d_\infty\left(\frac{Q}{P_{S'}}\right), \sqrt{\frac{\epsilon}{2} d_\infty\left(\frac{Q}{P_{S'}}\right)}\right\}.
\end{aligned}
$$

For convenience, define $d_\infty \coloneqq \sup_{S, S', Q \in \mathcal{Q}_{S,\mu}} d_\infty\left(\frac{Q}{P_{S'}}\right)$.

Thus, if we define $\beta \coloneqq \min\left\{\frac{\epsilon}{\sqrt{2\mu}} d_\infty, \sqrt{\frac{\epsilon}{2} d_\infty}\right\}$, then the family $\mathcal{Q}_{m,\mu}$ is $\beta$-uniformly stable. $\qquad\square$

### B.7 Proof of Lemma 6

**Lemma 6.** *Suppose* $\mathsf{D}_\infty(P_S \parallel P_{S'}) \leq \epsilon$ *for all* $S, S' \in \mathcal{Z}^m$ *differing by exactly one point. For some* $\mu \geq 0$, *define the sample-dependent set of distributions as* $\mathcal{Q}_{S,\mu} \coloneqq \{Q \colon \mathsf{D}(Q\|P_S) \leq \mu\}$, *and the corresponding family to be* $\mathcal{Q}_{m,\mu} = (\mathcal{Q}_{S,\mu})_{S \in \mathcal{Z}^m}$. *Then* $\mathcal{Q}_{m,\mu}$ *is* $\beta$-stable *for* $\beta = \min\left\{2\epsilon, \frac{\epsilon}{\sqrt{2\mu}}, \sqrt{\frac{\epsilon}{2}}\right\}$.

*Proof.* This follows from Lemmas 12 and 13. $\qquad\square$

**Lemma 12.** *If* $\mathsf{D}_\infty(P_S \parallel P_{S'}) \leq \epsilon$ *for all* $S, S' \in \mathcal{Z}^m$ *differing by exactly one point, then* $\mathcal{Q}_{m,\mu}$ *is* $\beta$-uniformly stable with $\beta = \min\left\{\frac{\epsilon}{\sqrt{2\mu}}, \sqrt{\frac{\epsilon}{2}}\right\}$.

*Proof.* Consider an arbitrary $Q \in \mathcal{Q}_{S,\mu}$.

Case (1): $\mathsf{D}(Q \parallel P_{S'}) \leq \mu$.
In this case, $Q \in \mathcal{Q}_{S',\mu}$, so we choose $Q' = Q$, and thus $\|Q' - Q\|_{\mathrm{TV}} = 0$.

Case (2): $\mathsf{D}(Q \parallel P_{S'}) > \mu$.
We consider $Q' = \lambda Q + (1 - \lambda)P_{S'}$, for $\lambda = \frac{\mathsf{D}(Q\|P_S)}{\mathsf{D}(Q\|P_{S'})} < 1$. We show that $Q' \in \mathcal{Q}_{S',\mu}$ as follows:

$$
\mathsf{D}(Q' \parallel P_{S'}) = \mathsf{D}(\lambda Q + (1-\lambda)P_{S'} \parallel P_{S'}) \leq \lambda \mathsf{D}(Q \parallel P_{S'}) + (1-\lambda)\mathsf{D}(P_{S'} \parallel P_{S'}) = \mathsf{D}(Q \parallel P_S) \leq \mu,
$$

where the inequality is by the convexity of relative entropy.

We can upper bound $\|Q' - Q\|_{\mathrm{TV}}$ in two different ways.
One way is to directly upper bound the TV distance as follows:

$$
\begin{aligned}
\|Q' - Q\|_{\mathrm{TV}} &= \|\lambda Q + (1 - \lambda)P_{S'} - Q\|_{\mathrm{TV}} \\
&= (1 - \lambda)\|Q - P_{S'}\|_{\mathrm{TV}} \\
&= \left[1 - \frac{\mathsf{D}(Q \parallel P_S)}{\mathsf{D}(Q \parallel P_{S'})}\right]\|Q - P_{S'}\|_{\mathrm{TV}} \\
&= \left[\mathsf{D}(Q \parallel P_{S'}) - \mathsf{D}(Q \parallel P_S)\right]\frac{\|Q - P_{S'}\|_{\mathrm{TV}}}{\mathsf{D}(Q \parallel P_{S'})} \\
&\leq \frac{\mathsf{D}(Q \parallel P_{S'}) - \mathsf{D}(Q \parallel P_S)}{\sqrt{2\mathsf{D}(Q \parallel P_{S'})}} \qquad\qquad \text{(Pinsker's inequality).}
\end{aligned}
$$

Alternatively, we can upper bound the TV distance by upper bounding the KL divergence as follows:

$$
\begin{aligned}
\mathsf{D}(Q \parallel Q') &= \mathsf{D}(Q \parallel \lambda Q + (1 - \lambda)P_{S'}) \\
&\leq (1 - \lambda)\mathsf{D}(Q \parallel P_{S'}) \qquad\qquad \text{(convexity of relative entropy)} \\
&= \left[1 - \frac{\mathsf{D}(Q \parallel P_S)}{\mathsf{D}(Q \parallel P_{S'})}\right]\mathsf{D}(Q \parallel P_{S'}) \\
&= \mathsf{D}(Q \parallel P_{S'}) - \mathsf{D}(Q \parallel P_S) \\
\implies \|Q' - Q\|_{\mathrm{TV}} &\leq \sqrt{\frac{\mathsf{D}(Q \parallel P_{S'}) - \mathsf{D}(Q \parallel P_S)}{2}} \qquad\qquad \text{(Pinsker's inequality).}
\end{aligned}
$$

We upper bound the common term $\mathsf{D}(Q \parallel P_{S'}) - \mathsf{D}(Q \parallel P_S)$ as follows:

$$
\begin{aligned}
\mathsf{D}(Q \parallel P_{S'}) - \mathsf{D}(Q \parallel P_S) &= \mathop{\mathbb{E}}_{h \sim Q}\left[\log \frac{Q(h)}{P_{S'}(h)}\right] - \mathop{\mathbb{E}}_{h \sim Q}\left[\log \frac{Q(h)}{P_S(h)}\right] \quad \text{(def. of relative entropy)} \\
&= \mathop{\mathbb{E}}_{h \sim Q}\left[\log \frac{P_S(h)}{P_{S'}(h)}\right] \\
&\leq \mathsf{D}_\infty(P_S \parallel P_{S'}).
\end{aligned}
$$

Putting this together, we obtain:

$$
\|Q' - Q\|_{\mathrm{TV}} \leq \frac{\mathsf{D}(Q \parallel P_{S'}) - \mathsf{D}(Q \parallel P_S)}{\sqrt{2\mathsf{D}(Q \parallel P_{S'})}} < \frac{\mathsf{D}_\infty(P_S \parallel P_{S'})}{\sqrt{2\mu}} \leq \frac{\epsilon}{\sqrt{2\mu}}.
$$

$$
\begin{aligned}
\|Q' - Q\|_{\mathrm{TV}} &\leq \min\left\{\frac{\mathsf{D}(Q \parallel P_{S'}) - \mathsf{D}(Q \parallel P_S)}{\sqrt{2\mathsf{D}(Q \parallel P_{S'})}}, \sqrt{\frac{\mathsf{D}(Q \parallel P_{S'}) - \mathsf{D}(Q \parallel P_S)}{2}}\right\} \\
&\leq \min\left\{\frac{\mathsf{D}_\infty(P_S \parallel P_{S'})}{\sqrt{2\mu}}, \sqrt{\frac{\mathsf{D}_\infty(P_S \parallel P_{S'})}{2}}\right\} \\
&\leq \min\left\{\frac{\epsilon}{\sqrt{2\mu}}, \sqrt{\frac{\epsilon}{2}}\right\}.
\end{aligned}
$$

So if we define $\beta := \min\left\{\frac{\epsilon}{\sqrt{2\mu}}, \sqrt{\frac{\epsilon}{2}}\right\}$, then the family $\mathcal{Q}_{m,\mu}$ is $\beta$-uniformly stable. $\qquad\square$

**Lemma 13.** *If* $\mathsf{D}_\infty(P_S \parallel P_{S'}) \leq \epsilon$ *for all* $S, S' \in \mathcal{Z}^m$ *differing by exactly one point, then* $\mathcal{Q}_{m,\mu}$ *is* $\beta$-uniformly stable with $\beta = 2\epsilon$.

*Proof.* For convenience, we measure stability using the total variation distance rather than $\ell_1$, and then present the final bound in terms of $\ell_1$ stability.

Consider an arbitrary $Q \in \mathcal{Q}_{S,\mu}$.

Case (1): $\mathsf{D}(Q \parallel P_{S'}) \le \mathsf{D}(Q \parallel P_S)$.
In this case, $Q \in \mathcal{Q}_{S',\mu}$, so we choose $Q' = Q$, and thus $\|Q' - Q\|_{\mathrm{TV}} = 0$.

Case (2): $\mathsf{D}(Q \parallel P_{S'}) > \mathsf{D}(Q \parallel P_S)$.
We consider $Q' = \lambda Q + (1 - \lambda) P_{S'}$, for $\lambda = \frac{\mathsf{D}(Q \parallel P_S)}{\mathsf{D}(Q \parallel P_{S'})} < 1$. We show that $Q' \in \mathcal{Q}_{S',\mu}$ as follows:

$$\mathsf{D}(Q' \parallel P_{S'}) = \mathsf{D}(\lambda Q + (1 - \lambda) P_{S'} \parallel P_{S'}) \le \lambda \mathsf{D}(Q \parallel P_{S'}) + (1 - \lambda) \mathsf{D}(P_{S'} \parallel P_{S'}) = \mathsf{D}(Q \parallel P_S) \le \mu,$$

where the inequality is by the convexity of relative entropy.

Next we will upper bound $\mathsf{D}(Q' \parallel P_S)$. For this we will use the fact that $\mathsf{D}(P_S \parallel P_{S'}) \le 2\epsilon^2$. This fact is from [Popescu et al.] and we provide an alternate proof in Lemma 14 below. Given the lemma we have

$$
\begin{aligned}
\mathsf{D}(Q \parallel P_{S'}) - \mathsf{D}(Q \parallel P_S) &= \mathop{\mathbb{E}}_{h \sim Q}\left[\log \frac{P_S(h)}{P_{S'}(h)}\right] \\
&= \mathop{\mathbb{E}}_{h \sim P}\left[\log \frac{P_S(h)}{P_{S'}(h)}\right] + \left(\mathop{\mathbb{E}}_{h \sim Q} - \mathop{\mathbb{E}}_{h \sim P}\right)\left[\log \frac{P_S(h)}{P_{S'}(h)}\right] \\
&\le \mathsf{D}(P_S, P_{S'}) + \epsilon \|Q - P\|_{\mathrm{TV}} \\
&\le 2\epsilon^2 + \epsilon \|Q - P_S\|_{\mathrm{TV}} \\
&\le 2\epsilon^2 + \epsilon \sqrt{\frac{\mathsf{D}(Q \parallel P_S)}{2}}. \quad \text{(Pinsker's inequality)} \quad (21)
\end{aligned}
$$

Next we show that $Q$ and $Q'$ are close in total variation distance. We consider two cases:

**Case a:** $\mathsf{D}(Q \parallel P_S) \le 2\epsilon^2$. Using convexity of $\mathsf{D}(Q \parallel .)$ we have

$$
\begin{aligned}
\mathsf{D}(Q \parallel Q') &\le (1 - \lambda) \mathsf{D}(Q \parallel P_{S'}) \\
&= \mathsf{D}(Q \parallel P_{S'}) - \mathsf{D}(Q \parallel P_S) \\
&\le 2\epsilon^2 + \epsilon \sqrt{\frac{\mathsf{D}(Q \parallel P_S)}{2}} \quad \text{[from (21)]} \\
&\le 3\epsilon^2.
\end{aligned}
$$

Using Pinsker's inequality we can conclude that $\|Q - Q'\|_{\mathrm{TV}} \le 2\epsilon$.

**Case b:** $\mathsf{D}(Q \parallel P_S) > 2\epsilon^2$. We have

$$
\begin{aligned}
\|Q - Q'\|_{\mathrm{TV}} &= (1 - \lambda) \|Q - P_{S'}\|_{\mathrm{TV}} \\
&= \big(\mathsf{D}(Q \parallel P_{S'}) - \mathsf{D}(Q \parallel P_S)\big) \frac{\|Q - P_{S'}\|_{\mathrm{TV}}}{\mathsf{D}(Q \parallel P_{S'})} \\
&\le \big(\mathsf{D}(Q \parallel P_{S'}) - \mathsf{D}(Q \parallel P_S)\big) \frac{1}{\sqrt{2\mathsf{D}(Q \parallel P_{S'})}} \\
&\quad \text{[ from Pinsker's inequality and the fact that } \mathsf{D}(Q \parallel P_{S'}) > \mathsf{D}(Q \parallel P_S) \text{]} \\
&\le \frac{2\epsilon^2}{\sqrt{2\mathsf{D}(Q \parallel P_{S'})}} + \frac{\epsilon}{2} \quad \text{[from (21)]} \\
&\le 2\epsilon \quad \text{[since } \mathsf{D}(Q \parallel P_S) > 2\epsilon^2 \text{].}
\end{aligned}
$$

$\square$

**Lemma 14.** *If* $\mathsf{D}_\infty(P_S, P_{S'}) \le \epsilon$ *for all* $S, S' \in \mathcal{Z}^m$ *differing by exactly one point, then* $\mathsf{D}(P_S \parallel P_{S'}) \le 2\epsilon^2$.

*Proof.* Suppose $\mathsf{D}_\infty(P_S, P_{S'}) \le \epsilon$ and $\mathsf{D}_\infty(P_{S'}, P_S) \le \epsilon$. Then,

$$
\begin{aligned}
\mathsf{D}(P_S \parallel P_{S'}) + \mathsf{D}(P_{S'} \parallel P_S) &= \mathop{\mathbb{E}}_{x \sim P_S}\left[\log \frac{P_S(x)}{P_{S'}(x)}\right] + \mathop{\mathbb{E}}_{x \sim P_{S'}}\left[\log \frac{P_{S'}(x)}{P_S(x)}\right] \\
&= \mathop{\mathbb{E}}_{x \sim P_S}\left[\log \frac{P_S(x)}{P_{S'}(x)} + \log \frac{P_{S'}(x)}{P_S(x)}\right] + \mathop{\mathbb{E}}_{x \sim P_{S'} - P_S}\left[\log \frac{P_{S'}(x)}{P_S(x)}\right] \\
&= \epsilon \sum_x \left|P_{S'}(x) - P_S(x)\right| \qquad (\text{since } \mathsf{D}_\infty(P_S, P_{S'}), \mathsf{D}_\infty(P_{S'}, P_S) \le \epsilon) \\
&= \epsilon \sum_{P_S(x) > 0} P_S(x)\left|\frac{P_{S'}(x)}{P_S(x)} - 1\right|. \qquad (P_S(x) = 0 \text{ implies } P_{S'}(x) = 0)
\end{aligned}
$$

Next, since both $\mathsf{D}_\infty(P_{S'}, P_S)$ and $\mathsf{D}_\infty(P_S, P_{S'})$ are bounded by $\epsilon$, we have

$$
\begin{aligned}
\left|\frac{P_{S'}(x)}{P_S(x)} - 1\right| &\le \max\left(e^\epsilon - 1, 1 - e^{-\epsilon}\right) \\
&\le e^\epsilon - 1.
\end{aligned}
$$

Hence we can conclude that

$$
\begin{aligned}
\mathsf{D}(P_S \parallel P_{S'}) + \mathsf{D}(P_{S'} \parallel P_S) &\le \epsilon(e^\epsilon - 1) \sum_{P_S(x) > 0} P_S(x) \\
&\le \epsilon(e^\epsilon - 1) \\
&\le 2\epsilon^2.
\end{aligned}
$$

$\square$

## B.8 Proof of Lemma 7

**Lemma 7.** *Suppose $\|P_S - P_{S'}\|_1 \le \epsilon$ for all $S, S' \in \mathcal{Z}^m$ differing by exactly one point. For some $\mu \ge 0$, define the sample-dependent set of distributions as $\mathcal{Q}_{S,\mu} := \{Q : \|Q - P_S\|_1 \le \mu\}$, and the corresponding family to be $\mathcal{Q}_{m,\mu} = (\mathcal{Q}_{S,\mu})_{S \in \mathcal{Z}^m}$. Then $\mathcal{Q}_{m,\mu}$ is $\beta$-stable for $\beta = \frac{\epsilon}{2}$.*

*Proof.* For convenience, we do the computations using the total variation distance rather than $\ell_1$.

Since $\|P_S - P_{S'}\|_{\mathrm{TV}} \le \frac{\epsilon}{2}$, there exists a coupling $C_1$ of $P_S$ and $P_{S'}$ such that if $(X, X') \sim C_1$, we have $\mathbb{P}[X \ne X'] \le \frac{\epsilon}{2}$. Similarly, since $\|P_S - Q\|_{\mathrm{TV}} \le \frac{\mu}{2}$, there exists a coupling $C_2$ of $P_S$ and $Q$ such that if $(X, Y) \sim C_2$, we have $\mathbb{P}[X \ne Y] \le \frac{\mu}{2}$. Now construct a coupling $C_3$ as follows. First, sample $X \sim P_S$. Then, sample $X' \sim C_1$ conditioned on $X$, and independently, sample $Y \sim C_2$ conditioned on $X$. Set

$$
Y' = \begin{cases} X' & \text{if } X = Y \\ Y & \text{otherwise.} \end{cases}
$$

Let $Q'$ be the distribution of $Y'$. Note that $\mathbb{P}[X = Y] \ge 1 - \frac{\mu}{2}$, so $\mathbb{P}[Y' = X'] \ge 1 - \frac{\mu}{2}$, which implies that $\|P_{S'} - Q'\|_{\mathrm{TV}} \le \frac{\mu}{2}$. Furthermore, by a union bound, we have

$$
\mathbb{P}[Y' = Y] = \frac{\mu}{2} + \mathbb{P}[X' = X = Y] \ge \frac{\mu}{2} + 1 - (\mathbb{P}[X \ne Y] + \mathbb{P}[X \ne X']) \ge \frac{\mu}{2} + 1 - \left(\frac{\mu}{2} + \frac{\epsilon}{2}\right) = 1 - \frac{\epsilon}{2}.
$$

So, $\|Q - Q'\|_{\mathrm{TV}} \le \frac{\epsilon}{2}$. $\square$