[Reviews · NeurIPS 2020]

Review 1

Summary and Contributions: The paper presents PAC Bayes bounds for the generalization error of classifiers. These bounds consider the sample-dependent prior framewrok, in which the prior depends on the particular training instances. The paper first introduces two general bounds that apply in every situation, each bound using a different divergence to measure distances between probability distributions. These bounds make no assumptions on the priors. Then the bounds are particularized for the case when the priors are stable in the sense that if two samples differ in exactly one instance, their priors are close. In this case the general bounds are further refined into non-trivial bounds. Finally, Theorem 4 introduces a framework to develop tight PAC-Bayes bounds with stable priors and with an extra requirement of a minimum probability for these priors.

Strengths: The paper addresses a very relevant problem for the PAC-Bayes community, the development of theoretical tools to study the generalization of neural networks with sample dependent priors. The theoretical results are strong and well developed, I think these results will open lines for its application on the characterization of neural networks. The results are very well presented and compared with the key papers they build on. In my opinion the contributions of this paper are significant and mean an advance in the study of sample-dependent priors.

Weaknesses: The main limitation of this work is the absence of empirical work. I think one of the strong points with PAC-Bayes theory is that it achieves non-trivial bounds, indeed very tight ones. Therefore, in my view, potential readers with a more applied interest would benefit from the introduction of an illustrative empirical section in which the bounds presented in this paper are used to analyze the generalization of actual classifiers. This empirical section could also include a comparison with [Dziugaite and Roy 2018] and/or other related works. Besides, I had to read several times the appendix and some of the hey related references to follow the details of the presentation of the results.

Correctness: The paper is theoretical, the proofs of all the results are well detailed in the appendix. I think they are correct but I couldn't check them vey carefully. There is no experimental work.

Clarity: The paper is well written. Just minor issues to correct: - lines 118, 121, a closing ")" is missing in the numerator of the log - line 125: The notions... are: - lines 148 and 154: Rademacher is misspelled - line 158 "be a family" is repeated -line 161: I think Q_{m,\mu} is a family of distributions on the hypothesis set, not a sample-dependent hypothesis set. Maybe the paper could give more detail about how to go from the Rademacher complexity defined for sets of hypothesis to the Rademacher complexity for the randomized classifier, as I think it is the case. In section 5 all Q are termed as distributions

Relation to Prior Work: In my view the background is well described and the contributions of this work are clearly discussed with related works.

Reproducibility: Yes

Additional Feedback: -------------------------------------------------------------------- Update after rebuttal: I thank the authors for their response to my comments. After reading the other reviews and the authors' response I have decided to not change my review.


Review 2

Summary and Contributions: This paper considers the problem that how to derive PAC-Bayes bounds with sample-dependent priors. First, with the concept of covering numbers, two general PAC-Bayes bounds with the sample-dependent priors are derived, based on which two non-trivial generalization bounds are obtained if the priors satisfy prior stability. Then, a framework for deriving PAC-Bayes bounds under prior stability is presented by the notation of hypothesis set stability, based on which a refined PAC-Bayes bound for infinite-Rényi divergence prior stability is derived. The main contribution of this paper is a solid theoretical framework, which can be used to derive PAC-Bayes bounds with sample-dependent priors if the prior satisfies certain stability condition and thus answers the question theoretically.

Strengths: This paper studies a significant problem in PAC-Bayesian learning and presents novel and solid theoretical contributions, which will yield a positive impact to the NeurIPS community.

Weaknesses: Although the contributions of this paper are theoretical and solid, it may be better to empirically evaluate the usefulness of the framework.

Correctness: All of the claims in the main text are sound.

Clarity: The paper is well written, and the results are clearly presented in the main text. I only find a typos and doubt one statement. * line 162, “$(\mathcal{Q})_{S\in\mathcal{Z}^m }$” ---> “$(\mathcal{Q}_S)_{S\in\mathcal{Z}^m }$”.

Relation to Prior Work: This paper discusses the related work in detail. It is clear that the contributions are different from that of the prior work.

Reproducibility: Yes

Additional Feedback: ------------------After rebuttal----------- Thank you for your response. I have also read other reviews and will not change my score.


Review 3

Summary and Contributions: POST-REBUTTAL UPDATE The author rebuttal did not answer my questions about the interplay between various choices (e.g., selecting epsilon in a data-dependent way, and the effect on the bounds). There were only promises to add some discussion, and so I trust the authors to do so. I was surprised to see the authors mention an Appendix C; this appendix contains novel results but was not referenced in the main text. I do not think it is fair to consider this appendix, given the later deadline for the supplementary material. For these two reasons, my score is unchanged. ----------------------------------------- This work develops novel, non-uniform-and-data-dependent generalization error bounds; most of the bounds are of a PAC-Bayesian type, while one of the results (Theorem 2) is for the standard "algorithms plays a single point hypothesis" type. The primary theme is to develop PAC-Bayesian bounds for the interesting situation where the prior is data-dependent (and hence no longer is strictly a prior in the statistical sense). The authors begin by (Theorem 2) developing an extension to a major result of Foster et al. (2019), swapping in the earlier upper bound a maximum transductive Rademacher complexity term for an expected transductive Rademacher complexity term. They then provide two results, each of which upper bounds the transductive Rademacher complexity (in the case when the hypothesis class is a particular, data-dependent class of probability distributions over the original hypothesis space H): (i) Lemma 1 bounds it in terms of the D_\infty (max divergence) covering numbers; this is later useful when making a connection to (e.g.) data-dependent priors that are chosen in a differentially private way (though DP technically isn't required) (ii) Lemma 2 bounds it in terms of the $\ell_1$-covering numbers and a type of Rademacher complexity of H. The above 3 results (plus some technical lemma) yield the first main result, a PAC-Bayesian bound allowing for data-dependent priors and which has certain types of covering numbers on the RHS (and also a type of Rademacher complexity of H in the case of the $\ell_1$ metric). The authors then show simplified versions of Theorem 1 in the cases when the prior has either low D_\infty sensitivity (doesn't change much when one example is swapped for another) or low ell_1 sensitivity. Another result (Theorem 3), for beta-uniformly stable algorithms, apparently improves on a result of Foster et al. (2019). An application (Theorem 4) is given for priors with low D_\infty sensitivity.

Strengths: The most clear strengths of this work are its extensions/strengthenings of the results of Foster et al. (2019). It is difficult to speak much about significance, because the authors don't spend much time talking about applications. The first, Theorem 2 (extending Theorem 1 of Foster et al. (2019)), is actually a small contribution in my opinion; the proof is a minor variation of Theorem 1 of Foster et al. (2019), using a bounded differences inquality to relate the transductive Rademacher complexity to its expectation. I have a question for the authors: wouldn't it be more useful to give a fully empirical version which does not involving taking the expectation WRT U? And can't this be done by again using the bounded differences inequality, in the same way that we do so when getting fully observable generalization error bounds with standard Rademacher complexity? The second extension/strengthening is Theorem 3, which the authors say is tigher than an analogous result of Foster et al. (2019). However, the authors should mention which result they are talking about (I took a guess that you meant Theorem 2 of Foster et al.). While I did not closely look at the proof, based on how it starts, I suppose what is meant is that this result (Theorem 3) is tighter because of (as the authors say in the appendix) "the special structure in our setting". It would have been useful to comment on this in the main text, explaining what you mean. Theorem 4 could be an interesting result, and it was mentioned that it is superficially similar to one of the results of Dziugate and Roy (2018a). I would say this result is *potentially* significant, but I cannot confidently conclude significance (in as much can be done in the review process, anyhow) given the discussion (or lack thereof) of the result.

Weaknesses: There are two major weaknesses; (I) and (II). My current score is mainly but not entirely due to (I) (I) A major weakness of this work is that it tries to pack in too many results (including some lemmas and a long proof), at the expense of a discussion of the results (including their utility) and showing applications. As a paper that tries to extend results of Foster et al. (2019) and provide complementary or competing results to Dziugate and Roy's (2018a) results, I found it odd that applications weren't given. Foster et al. (2019) provided several applications in a dedicated section, and Dziugate and Roy did give some experiments. The authors' work would be much stronger if they could show some (theoretical) applications. There is a complicated interplay between various choices; here are some natural questions: - how might we select epsilon in a sample-dependent way and what are the effects on the final bounds, knowing that changing epsilon can have implications for the data-dependent prior that is selected; - what algorithm(s) should we have in mind (along with a mechanism for selecting the prior [which itself would come along with some choice of epsilon or beta]); is there a non-zero value of epsilon for which we win something (i.e., do better than the case of epsilon=0 where we have a data-independent prior). Giving some more understanding for how the various covering numbers behave as well would be useful; many of the covering numbers appearing in this work are non-standard. If they were standard, perhaps you would have citations; as they don't seem to be, giving some examples or discussion is in order. Note that you can save about a page by moving a long proof to the appendix, enabling more discussion/applications/examples and thereby increasing the impact of this paper. (II) In Theorem 4, I am somehow quite uneasy about the assumption P_S(h) >= eta > 0. This implies that the data-dependent prior has support upper bounded by 1/eta, and so, once the prior is fixed, we are essentially in a finite-class situation (as a posterior that places mass where the prior is zero will make the KL blow up). Isn't this a serious issue? If not, why not? Please do respond to this point in your rebuttal; I really would appreciate a clarification.

Correctness: I have some confidence in the correctness of the proofs/results, but I admit I did not look through the proofs in the appendix. The techniques seem to be more or less standard (though some techniques are just from last year, I admit); their synthesis is the novelty, I believe.

Clarity: Leaving aside my point about insufficient discussion and examples, I found the paper to be clear, except for one thing. On line 211, the definition of $\mathcal{Q}_m$ is not clear. As written, it appears to be a sequence. I believe what is actually meant (based on the appendix) is that it is a union of sets (each $S$ giving rise to one set). Also, the LHS of equation (6) hasn't been defined. The definition is only given in the math display between lines 403 and 404 in the appendix.

Relation to Prior Work: I believe all the relevant prior work was cited. As mentioned earlier, some discussion contrasting Theorem 4 with the results of Dziugate and Roy (2018a) would be helpful. If your results are totally of a different nature, it should be easy to say why. If not, it's interesting to say how they differ.

Reproducibility: Yes

Additional Feedback: The proof of Theorem 2 (in the appendix) seems to largely use the proof of Theorem 1 of Foster et al. (2019), but this is not clearly stated. Please fix up your proof so that the steps are clear (specifically, on line 421, I suspect "the proof" is referring to the proof of Theorem 1 of Foster et al. (2019)).

[Author Response · NeurIPS 2020]

**High-level comments.**

We thank the reviewers for their thoughtful feedback. The main focus of our paper has been an extensive theoretical analysis of this rich research area. This required tackling a variety of different questions and deriving a number of new results in order to handle sample-dependent priors with reasonable generality. We agree that our theory can and should be developed into applications, and this is the focus of ongoing work.

**Reviewer 4.**

- **Main limitation:** Please see high-level comments above. We will also work on improving the readability of the final version of the paper.

- **Typos:** We thank the reviewer for catching them (lines 118, 121, 125, 148, 154, 158) and will fix them all.

- **Line 161:** We agree that "the paper could give more detail about how to go from the Rademacher complexity defined for sets of hypothesis to the Rademacher complexity for the randomized classifier" and will make this more explicit.

**Reviewer 6.**

- **Weakness:** Please see high-level comments above.

- **Line 162:** We agree that line 162 is missing a subscript ($S$, as the reviewer pointed out, plus $\mu$).

- **Broader Impact:** We will also expand on the Broader Impact section.

**Reviewer 7.**

- **Weakness I:** Please see high-level comments above. Additionally, we thank the reviewer for the suggestion about adding some theoretical applications as in Foster et al. (2019). In our final version, we will initiate a discussion of the choice of the parameters (and their implications) and seek to outline a theoretical application along these lines.

- **Weakness II (Theorem 4):** We agree with the reviewer. Indeed, this implies a finite class, although the bounds would be non-trivial even for a fairly large class, since the dependence on $1/\eta$ is only logarithmic. Motivated precisely by this concern, in Appendix C, we present an alternative bound without any assumption on the minimum probability, at the price of a slightly worse dependence on $\epsilon$. On the other hand, note that Theorem 4 already yields a non-trivial generalization bound based on the observation that $|\mathcal{H}| \leq 1/\eta$. This means that $\mathfrak{R}_m(\mathcal{H}) \leq \sqrt{\frac{2\log(1/\eta)}{m}}$, and hence Theorem 4 yields a generalization bound scaling as $O\left(\frac{1}{\sqrt{m}}\right)$ even for a somewhat modest $\epsilon = 1/\sqrt{m}$, comparable to Dziugaite and Roy (2018a).

- **Empirical transductive Rademacher complexity in Theorem 2:** That is a good suggestion. A version of the theorem with an empirical transductive Rademacher complexity would be more useful in practice and we will include that in the final version. That is straightforward to derive using the fact that the empirical transductive Rademacher sharply concentrates around its expectation.

- **Clarifications for Theorem 3:** We will move these clarifications from the appendix (in the proof) to the main text and further explain the special structure, which is that the loss function in our setting, $Q \mapsto \langle Q, \ell \rangle$, is *linear*. This linearity yields the bounds in the displayed equations after line 497.

- **Discussion of Theorem 4:** As described in the Related Work section, the results of Dziugaite and Roy (2018a) are for a completely different setting than ours. In the final version, we will give a more explicit discussion of Theorem 4 and its implications, specifically contrasting Theorem 4 with the results of Dziugaite and Roy (2018a).

- **Clarification for $\mathcal{Q}_m$ and $\mathfrak{R}_m^\diamond(\mathcal{Q}_m)$:** $\mathcal{Q}_m$ is a family of sets of distributions (lines 202-203). This is in contrast with $\overline{\mathcal{Q}}_{U,m,\mu}$, which is a union (line 398, Appendix). Equation (6) is actually the definition of $\mathfrak{R}_m^\diamond(\mathcal{Q}_m)$ (lines 210-211).

- **Proof of Theorem 2:** We will revise the proof of Theorem 2 in the appendix so that all steps are more clear.

[Meta-Review · NeurIPS 2020]

This paper derives new PAC-Bayesian risk bounds for sample-dependent priors -- that is, priors that depend on the training data, which violates the classical PAC-Bayes setting. There has been a recent surge in papers about informed priors, and this paper takes an interesting step in this direction. Notably, it improves on recent work by Foster et al. (2019) on "hypothesis set stability," which incorporates ideas from transductive Rademacher complexity. The resulting risk bound makes no assumptions about the prior; the prior is characterized by covering numbers. The paper applies the bound to a family of sample-dependent priors that obeys a sensitivity condition (similar to algorithmic stability but designed for distributions). PAC-Bayes with informed priors has had a resurgence because it has been shown to yield non-vacuous risk bounds for neural networks (see, e.g., work by Dziugaite & Roy). Thus, the paper addresses a relevant, timely problem. The reviewers agree that the paper is solid; the results are new, interesting, and presented well. The main criticism is that the paper lacks an empirical investigation. The reason for using PAC-Bayes and data-dependent priors is to obtain bounds that are meaningful in practice. Papers on generalization bounds (especially for deep learning) have started to include experiments to demonstrate that (a) the bounds are non-vacuous and (b) tighter than others. Not including an empirical study is a big gap here; including one would have taken the paper to another level. I strongly encourage the authors to incorporate feedback from the reviewers into the paper -- especially from R7.